

# DCore: Integrated DMFT software for correlated electrons

Hiroshi Shinaoka[1*], Junya Otsuki[2], Mitsuaki Kawamura[3],
Nayuta Takemori[2] and Kazuyoshi Yoshimi[3]

**1** Department of Physics, Saitama University, Saitama 338-8570, Japan
**2** Research Institute for Interdisciplinary Science, Okayama University,
Okayama 700-8530, Japan
**3** Institute for Solid State Physics, University of Tokyo, Chiba 277-8581, Japan

★ shinaoka@mail.saitama-u.ac.jp

## Abstract

We present a new open-source program, DCore, that implements dynamical mean-field theory (DMFT). DCore features a user-friendly interface based on text and HDF5 files. It allows DMFT calculations of tight-binding models to be performed on predefined lattices as well as *ab initio* models constructed by external density functional theory codes through the Wannier90 package. Furthermore, DCore provides interfaces to many advanced quantum impurity solvers such as quantum Monte Carlo and exact diagonalization solvers. This paper details the structure and usage of DCore and shows some applications.

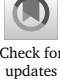

# 1  Introduction

Dynamical mean-field theory (DMFT) has become a standard theoretical tool for studying strongly correlated electronic systems [1]. In a DMFT calculation, an original lattice model is mapped to an effective Anderson impurity problem whose bath degrees of freedom are self-consistently determined. Although the DMFT formalism was originally proposed for models such as Hubbard models, it can be combined with density functional theory (DFT) based on *ab initio* calculations to describe the electronic properties of strongly correlated materials [2,3]. This composite framework, called DFT+DMFT, has been applied to various types of materials, such as cuprates [4], Fe-based superconductors [5–7], and $f$-electron materials [8–10].

To make DFT+DMFT available to more users and to promote the development of a community of users and developers, there is a strong need for an open-source program package with a user-friendly interface. Some open-source software packages for performing DFT+DMFT calculations have been developed. The TRIQS (Toolbox for Research on Interacting Quantum Systems) project [11] aims to provide the necessary components to implement DFT+DMFT codes, including a quantum Monte Carlo (QMC) impurity solver [12] and an interface to DFT codes [13]. The user can implement their own DMFT code using the building blocks in Python. The ALPS project provides a DMFT code for simple Hubbard models [14] and implementations of several continuous-time QMC (CT-QMC) impurity solvers [15–17]. w2dynamics [18] provides a state-of-the-art implementation of a QMC algorithm and includes a Python program for performing simple DFT+DMFT calculations. DMFTwDFT features a user-friendly interface and has interfaces to various DFT codes and the CT-QMC impurity solver [19]. DFT+embedded DMFT Functional (eDMFT) [20] features a full set of DFT+DMFT functionalities but it is not open-source software.

DCore is an open-source program package that implements (DFT+)DMFT calculations for multi-orbital systems. This package is built on top of existing TRIQS Python libraries and provides interfaces to external impurity solvers from the ALPS [15,16] and TRIQS projects [12,21] as well as exact-diagonalization solvers based on pomerol [22] and HΦ [23]. DCore provides

a flexible interface based on text and HDF5 files and does not require extensive expert knowledge. It also provides a set of command line tools to analyze a self-consistent solution and evaluate physical quantities such as momentum-resolved spectral functions.

The remainder of this paper is organized as follows. In Sec. 2, the methods used for implementing DCore are described. In Sec. 3, the structure of DCore and a flowchart of simulations are shown. In Sec. 4, the formats of input and output files are explained. In Sec. 5, brief instructions on the installation of DCore are provided. In Sec. 6, several run examples of DCore from simple Hubbard models to DFT+DMFT calculations for a correlated material are demonstrated. In Sec. 7, a summary and the conclusions of this paper are given.

## 2 Methodology

In this section, a brief explanation of the methodology used for implementing DCore is provided. For more technical details of the DFT+DMFT method, we refer the reader to a review article [2]. The current version of DCore implements only one-shot DFT+DMFT calculations based on Wannier90[1]. DCore does not support charge self-consistent calculations [9], which are supported by other programs such as eDMFT and DMFTwDFT. In a charge self-consistent calculation, the DFT effective potential is updated using the density matrix obtained by DMFT calculations. This is essential especially in performing lattice optimization. DCore specializes in the analysis of multi-orbital Hubbard models.

### 2.1 One-shot DFT+DMFT calculations and tight-bonding models

In one-shot DFT+DMFT calculations, one first performs "band calculations" usually using the local density approximation (LDA), where correlation effects in strongly correlated orbitals (e.g., partially filled $d$ orbitals) are not described accurately. One projects the effective one-body LDA Hamiltonian onto some tight-binding basis, which yields a multi-orbital tight-binding model. Then, one introduces local electron interactions for correlated orbitals into the tight-binding model. The resultant multi-orbital Hubbard model is solved using DMFT. We will discuss how to subtract the contributions of electron correlations at the LDA level from DMFT results (double-counting corrections) in a later section. One may not introduce additional interactions at the DMFT level for weakly correlated orbital for which LDA is accurate enough (e.g., deep oxygen orbitals).

### 2.2 Model

DCore is designed to solve a tight-binding model with periodic boundary conditions in second quantization. We denote the primitive vectors of the lattice by $\boldsymbol{a}_1$, $\boldsymbol{a}_2$, $\boldsymbol{a}_3$. Then, the coordinates of a unit cell can be specified by an integer array of length 3, $\boldsymbol{R} = (R_1, R_2, R_3)$, as $\sum_{i=1}^{3} R_i \boldsymbol{a}_i$. Each unit shell involves shells, which are sets of spin orbitals. We define a creation operator and an annihilation operator for the $\alpha$-th spin orbital in the $s$-th shell as $c_{\boldsymbol{R}s\alpha}^{\dagger}$ and $c_{\boldsymbol{R}s\alpha}$, respectively. In the reciprocal space, we define the corresponding creation and annihilation operators by

$$c_{\boldsymbol{k}s\alpha}^{\dagger} = \frac{1}{\sqrt{N}} \sum_{\boldsymbol{R}} e^{i\boldsymbol{k}\cdot\boldsymbol{R}} c_{\boldsymbol{R}s\alpha}^{\dagger}, \tag{1}$$

$$c_{\boldsymbol{k}s\alpha} = \frac{1}{\sqrt{N}} \sum_{\boldsymbol{R}} e^{-i\boldsymbol{k}\cdot\boldsymbol{R}} c_{\boldsymbol{R}s\alpha}, \tag{2}$$

---

[1]DCore uses a Wannier90 converter in TRIQS/DFTTools [13].

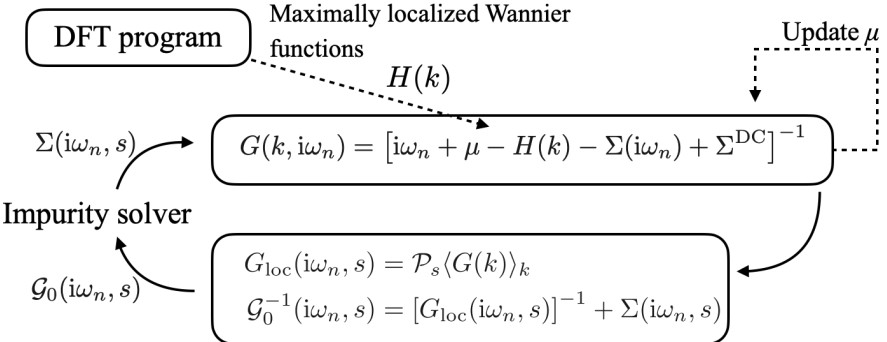

Figure 1: Illustration of the DMFT self-consistency cycle defined in Eqs. (6)–(10). The most numerically expensive step is the solution of the impurity problem, i.e., the calculation of the self-energy for a given $\mathcal{G}_0^{-1}$. $\langle\ldots\rangle_k$ denotes an average over the momentum space and $\mu$ is the chemical potential.

where momentum index $\boldsymbol{k} = (k_1, k_2, k_3)$ and $k_i \in \mathbb{Z}$. $N$ denotes the number of $\boldsymbol{k}$ points.

The Hamiltonian of a tight-binding model can be written as follows:

$$\mathcal{H} = \sum_{\boldsymbol{k}=1}^{N} \sum_{s,s'\in\text{all shells}} \sum_{\alpha\beta} H_{ss'}^{\alpha\beta}(\boldsymbol{k}) c_{\boldsymbol{k}s\alpha}^{\dagger} c_{\boldsymbol{k}s'\beta} + \sum_{\boldsymbol{R},s\in\text{crsh}} \mathcal{H}_{\text{int}}(\boldsymbol{R}, s), \tag{3}$$

$$= \sum_{\boldsymbol{R},\boldsymbol{R}'}^{N} \sum_{s,s'\in\text{all shells}} \sum_{\alpha\beta} H_{ss'}^{\alpha\beta}(\boldsymbol{R}-\boldsymbol{R}') c_{\boldsymbol{R}s\alpha}^{\dagger} c_{\boldsymbol{R}'s'\beta} + \sum_{\boldsymbol{R},s\in\text{crsh}} \mathcal{H}_{\text{int}}(\boldsymbol{R}, s), \tag{4}$$

where the first term in Eq. (3) denotes a non-interacting Hamiltonian. In the above equations, $s$ and $s'$ run either over correlated (crsh) "shells" or over all "shells" (i.e., the correlated shells and a non-interacting shell[2]). Hereafter, the number of the correlated shells is denoted by ncor. $\alpha$ and $\beta$ denote the spin-orbital index in each shell. The second term $\mathcal{H}_{\text{int}}(\boldsymbol{R}, s)$ denotes an interaction at the $s$-th correlated shell. This Hamiltonian is given as

$$\mathcal{H}_{\text{int}}(\boldsymbol{R}, s) = \frac{1}{2} \sum_{\alpha\beta\gamma\delta} U_{\alpha\beta\gamma\delta}(s) c_{\boldsymbol{R}s\alpha}^{\dagger} c_{\boldsymbol{R}s\beta}^{\dagger} c_{\boldsymbol{R}s\delta} c_{\boldsymbol{R}s\gamma}, \tag{5}$$

where $U_{\alpha\beta\gamma\delta}(s)$ denotes a spin-full four-rank Coulomb tensor. Note that $\mathcal{H}_{\text{int}}$ is limited to intra-shell interactions and local interactions in each unit cell.

## 2.3 DMFT formalism

DCore solves the following DMFT self-consistent equations, where the lattice Green's function $G$, the local Green's function $G_{\text{loc}}$, the noninteracting impurity-model Green's function $\mathcal{G}_0$, and the impurity/lattice self-energy $\Sigma^{\text{imp}}/\Sigma^{\text{lattice}}$ are defined in the spin-orbital space:

$$G(k, i\omega_n) = \left[i\omega_n + \mu - H(k) - \Sigma(i\omega_n) + \Sigma^{\text{DC}}\right]^{-1}, \tag{6}$$

$$G_{\text{loc}}(i\omega_n, s) = \mathcal{P}_s \langle G(k) \rangle_k \ (\forall s \in \text{crsh}), \tag{7}$$

$$\mathcal{G}_0^{-1}(i\omega_n, s) = \left[G_{\text{loc}}(i\omega_n, s)\right]^{-1} + \Sigma(i\omega_n, s) \ (\forall s \in \text{crsh}), \tag{8}$$

$$\Sigma_{\sigma\sigma'}^{\text{imp}}(i\omega_n, s) \leftarrow \mathcal{G}_0(i\omega_n, s), \ U_{\alpha\beta\gamma\delta}(s) \ (\forall s \in \text{crsh}; \sigma, \sigma' \in \{\uparrow, \downarrow\}), \tag{9}$$

$$\Sigma_{\sigma\sigma'}(i\omega_n, s) \leftarrow \Sigma_{\sigma\sigma'}^{\text{imp}}(i\omega_n, s). \tag{10}$$

---

[2]The term "non-interacting shell" means that there are no additional electron interactions introduced at the DMFT level for the shell.

Here, $\omega_n \equiv (2n+1)\pi T$ is a fermionic Matsubara frequency at temperature $T$, $\langle\cdots\rangle_k$ denotes an average over the momentum space, $s$ indexes correlated shells, and $\mathcal{P}_s$ is the projector to the $s$-th correlated shell. Double-counting corrections are denoted by $\Sigma^{DC}$ in Eq. (6) [see Eq. (15)]. Equations (6)–(10) are illustrated in Fig. 1.

All Green's functions and self-energies in Eqs. (6)–(10) are matrices at each Matsubara frequency, and their inversions are regarded as inverse matrices. The full self-energy $\Sigma(i\omega_n)$ in Eq. (6) is defined for all spin and shell components, and has the spin-block structure

$$\Sigma(i\omega_n) = \begin{pmatrix} \Sigma_{\uparrow\uparrow}(i\omega_n) & \Sigma_{\uparrow\downarrow}(i\omega_n) \\ \Sigma_{\downarrow\uparrow}(i\omega_n) & \Sigma_{\downarrow\downarrow}(i\omega_n) \end{pmatrix}, \tag{11}$$

where each spin-component has the internal shell structure

$$\Sigma_{\sigma\sigma'}(i\omega_n) = \begin{pmatrix} \Sigma_{\sigma\sigma'}(i\omega_n, s=0) & 0 & \cdots & 0 \\ 0 & \ddots & & \vdots \\ \vdots & & \Sigma_{\sigma\sigma'}(i\omega_n, s=\texttt{ncor}-1) & 0 \\ 0 & \cdots & 0 & 0 \end{pmatrix}, \tag{12}$$

where the last column and row correspond to the non-interacting shell. The block structures of the Green's functions are determined consistently with those of the self-energies in Eqs. (6)–(10).

The Green's functions and the self-energies are assumed to be either spin-diagonal or dense matrices in the spin space with both diagonal and off-diagonal components. The former is sufficient for computing a paramagnetic state or a collinear magnetic state polarized along the $z$ axis. The latter allows $\Sigma(i\omega_n)$, $G(i\omega_n)$, and $\mathcal{H}(k)$ to have spin-off-diagonal elements. This must be used when $\mathcal{H}$ has spin-orbit coupling that mixes the spin-up/down sectors or when one computes non-collinear magnetic states.

As illustrated in Fig. 2, by default, all correlated shells are assumed to be inequivalent shells that can have different self-energies. As described in the following sections, the user can assume that several correlated shells have the same self-energy. In this case, these correlated shells define one inequivalent shell (see Case 2 in Fig. 2).

The chemical potential $\mu$ is either adjusted to obtain a desired number of electrons or is fixed at a given value. In the former case, the value of $\mu$ is determined by using a bisection algorithm, which requires solving Eq. (6) multiple times at each self-consistent iteration. Note that the number of electrons per unit cell is given by $-\mathrm{Tr}\, G(\tau = \beta - 0^+)$, where $G(\tau) \equiv \beta^{-1} \sum_{n=-\infty}^{\infty} e^{-i\omega_n \tau} \langle G(k, i\omega_n)\rangle_k$ ($\beta$ and Tr denote inverse temperature and matrix trace, respectively).

In Eq. (10), the new self-energy is computed by solving an impurity model for each inequivalent shell. The non-interacting and interacting parts of the impurity model are defined by $\mathcal{G}_0(i\omega_n, s)$ and $U_{\alpha\beta\gamma\delta}(s)$, respectively. Some QMC impurity solvers accept the hybridization function defined below as an input:

$$\Delta(i\omega_n, s) = i\omega_n + \mu - \mathcal{P}_s \langle H(k)\rangle_k - \mathcal{G}_0^{-1}(i\omega_n, s). \tag{13}$$

After computing impurity self-energies $\Sigma^{\mathrm{imp}}(i\omega_n)$ for all inequivalent shells, we update $\Sigma(i\omega_n, s)$ by using simple linear mixing as

$$\Sigma^{\mathrm{new}}(i\omega_n, s) \leftarrow \sigma_{\mathrm{mix}} \Sigma^{\mathrm{imp}}(i\omega_n, s) + (1 - \sigma_{\mathrm{mix}}) \Sigma^{\mathrm{old}}(i\omega_n, s) \tag{14}$$

and go back to Eq. (6). Here, $\sigma_{\mathrm{mix}}$ ($0 < \sigma_{\mathrm{mix}} \leq 1$) is a mixing parameter.

DCore implements three different algorithms for computing the double-counting correction $\Sigma^{DC}$:

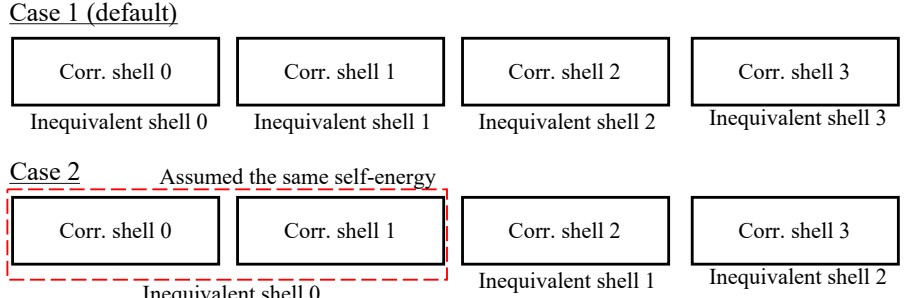

Figure 2: Correlated shells and inequivalent shells

- HF_DFT (default): Hartree-Fock (HF) contribution is subtracted from the DFT input.

- HF_imp: HF contribution computed from the local Green's function $G_{\text{loc}}$ [Eq. (7)] is subtracted from the impurity self-energy.

- FLL: A formula called fully-localized limit [24] is used.

For the default algorithm (HF_DFT), the double-counting correction is computed as follows:

$$\Sigma_{\alpha\beta}^{\text{DC}}(s) = \sum_{\gamma\delta} \left( U_{\alpha\gamma\beta\delta}(s) - U_{\alpha\gamma\delta\beta}(s) \right) \langle c_{s\gamma}^{\dagger} c_{s\delta} \rangle_0 \,, \tag{15}$$

where $\langle \cdots \rangle_0$ indicates the expectation value at the initial (Kohn-Sham) state with $k$-summation included, and $\{\alpha, \beta, \gamma, \delta\}$ index spin orbitals.

## 2.4 MPI parallelization

DCore uses MPI parallelization for performing numerically expensive parts of DMFT calculations. For instance, $k$-average of $G(k, i\omega_n)$ in Eq. (6) is executed in parallel. Besides, some impurity solvers, such as CT-QMC solvers, may use MPI parallelization. As described in the following sections, all programs in DCore are launched as a single process. Some of them internally invoke MPI processes. For some examples, please refer to Sec. 6.

## 3 Structure of DCore and flowchart of simulation

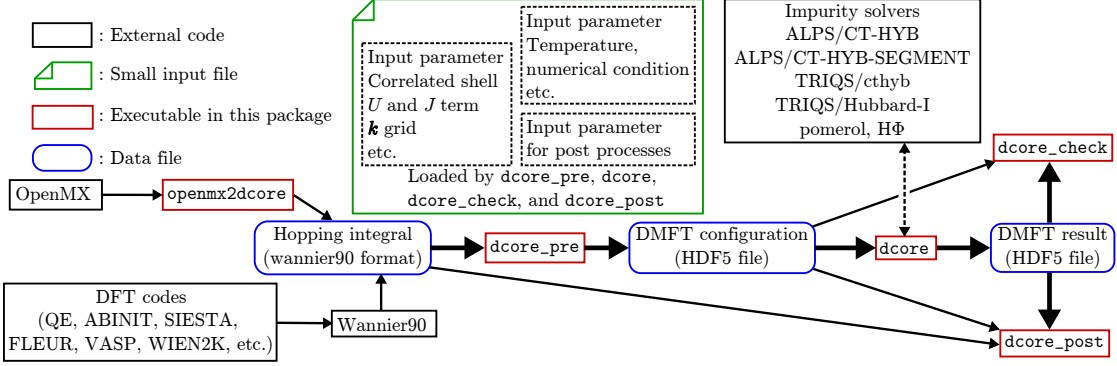

Figure 3: Structure of DCore.

DCore contains a set of programs that perform DMFT calculations for models and materials. The structure of the programs and data flow is summarized in Fig. 3.

DCore consists of four layers: (i) interface layer, (ii) DMFT self-consistent loop, (iii) convergence check and (iv) post-processing. These are respectively performed by the executables `dcore_pre`, `dcore`, `dcore_check` and `dcore_post`. Input parameters are provided by a single text file, which is read by all three programs. Data generated by `dcore_pre` and `dcore` are separately stored in an HDF5 file and passed to the next process.

(i) Interface layer: `dcore_pre`
`dcore_pre` generates the HDF5 file necessary for the DMFT loop. Users specify parameters that define a model, such as the hopping parameters on a certain lattice, and interactions. The hopping parameters are given either for preset models (e.g., square lattice, Bethe lattice) or using the Wannier90 format.

(ii) DMFT self-consistent loop: `dcore`
`dcore` is the main program for the DMFT calculations. The effective impurity problem is solved repeatedly to satisfy the self-consistency condition of the DMFT. For solving the impurity problem, dcore calls an external program. As an external program, we can select ALPS/CT-HYB [15], ALPS/CT-HYB-SEGMENT [16], or TRIQS/cthyb [12] as a CT-QMC solver, TRIQS/Hubbard-I [21] as a Hubbard-I solver, or pomerol [22] or HΦ [23] as an exact diagonalization (ED) solver.

(iii) Convergence check: `dcore_check`
`dcore_check` reads the results of self-consistent calculations (e.g., the self-energy) from the output HDF5 file of `dcore` and outputs several figures that are useful for checking the convergence of the results.

(iv) Post-processing: `dcore_post`
`dcore_post` computes some physical quantities from the converged solution of the DMFT loop. Currently, the (projected) density of states $A(\omega)$ and correlated band structures (momentum-resolved single-particle excitation spectrum) $A(\boldsymbol{k}, \omega)$ can be calculated. The analytic continuation of the self-energy from the Matsubara frequencies to real frequencies is performed using the Padé approximation.

# 4 Format of input and output files

In this section, the format of input and output files for DCore is briefly explained.

## 4.1 Input file format

The input file of DCore consists of six parameter blocks, respectively named `[model]`, `[system]`, `[impurity_solver]`, `[control]`, `[tool]`, and `[mpi]`. All the programs usually read the same file, but that some blocks are, even if present, irrelevant to some of the programs (see Table 1). Each block is described below.

1. `[model]` block
This block includes parameters for defining a model to be solved. The parameter types are divided into four parts: (i) Basic parameters, (ii) Lattice parameters, (iii) Interaction parameters, and (iv) Local potential parameters. In the following, we describe the parameters defined in each part.

Table 1: Blocks to be used for each program in DCore.

| Block | dcore_pre | dcore | dcore_check | dcore_post |
|---|---|---|---|---|
| model | Yes | Yes | Yes | Yes |
| system | - | Yes | Yes | Yes |
| impurity_solver | - | Yes | - | Yes |
| control | - | Yes | - | - |
| tool | - | - | Yes | Yes |
| mpi | - | Yes | - | Yes |

(i) Basic parameters

There are four basic parameters, namely `seedname`, `nelec`, `norb`, and `spin_orbit`. `seedname` determines the name of the model HDF5 file. `nelec` and `norb` specify the number of electrons per unit cell and the number of orbitals, respectively. `spin_orbit` specifies whether the model has spin-orbit interactions.

(ii) Lattice parameters

In DCore, `chain`, `square`, `cubic`, and `bethe` lattices are prepared as predefined models, as shown in Fig. 4. We can select a lattice type using the `lattice` parameter and set values of transfer integrals using parameters `t` and `t'`. For treating more realistic cases, such as DFT+DMFT calculations, hopping parameters in the Wannier90 format can be imported by selecting `wannier90`. In this mode, we can set a number of correlated shells in a unit cell and a mapping from correlated shells to equivalent shells using the `ncor` and `corr_to_inequiv` parameters, respectively. Figure 5 shows a schematic diagram of the shell structure in `wannier90` mode. For experts, the lattice data can be custom prepared using the `external` mode. In this mode, all necessary data should be directly made in the `dft_input` group of the model HDF5 file. For details, see the reference manual of DFTTools [13]. The information of reciprocal lattice vectors and the number of wave vectors can be also specified.

s (iii) Interaction parameters

The interaction part of the Hamiltonian is defined as Eq. (5). The interaction matrix $U_{\alpha\beta\gamma\delta}(i)$ is specified by the parameter `interaction`. In DCore, three types of `interaction`, namely (a) `kanamori`, (b) `slater_f`, and (c) `slater_uj`, are defined.

(a) If `interaction = kanamori`, the Kanamori-type interaction is used; i.e., $U_{(a\sigma_a)(b\sigma_b)(c\sigma_c)(d\sigma_d)} = V_{abcd}\delta_{\sigma_a\sigma_c}\delta_{\sigma_b\sigma_d}$ with $V_{aaaa} = U, V_{abab} = U', V_{abba} = J, V_{aabb} = J$, where $a \neq b$ ($a, b, c, d$ are orbital indices, and $\sigma_a, \sigma_b, \sigma_c, \sigma_d$ are spin indices). These

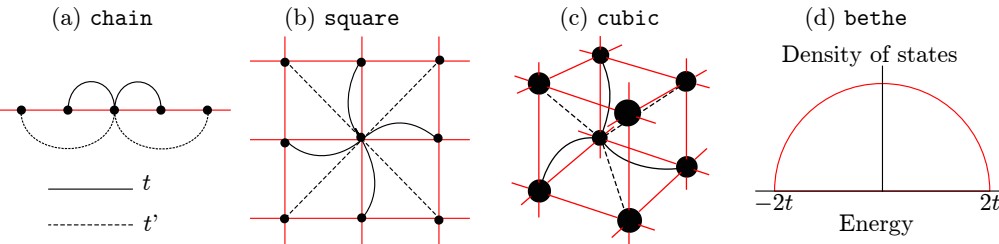

Figure 4: Predefined lattices in DCore. In the (a) `chain`, (b) `square`, and (c) `cubic` lattices, the solid (dotted) bonds indicate the transfer integrals $t$ ($t'$). For the (d) `bethe` lattice, a semicircular density of states with energy ranges $[-2t{:}2t]$ is defined.

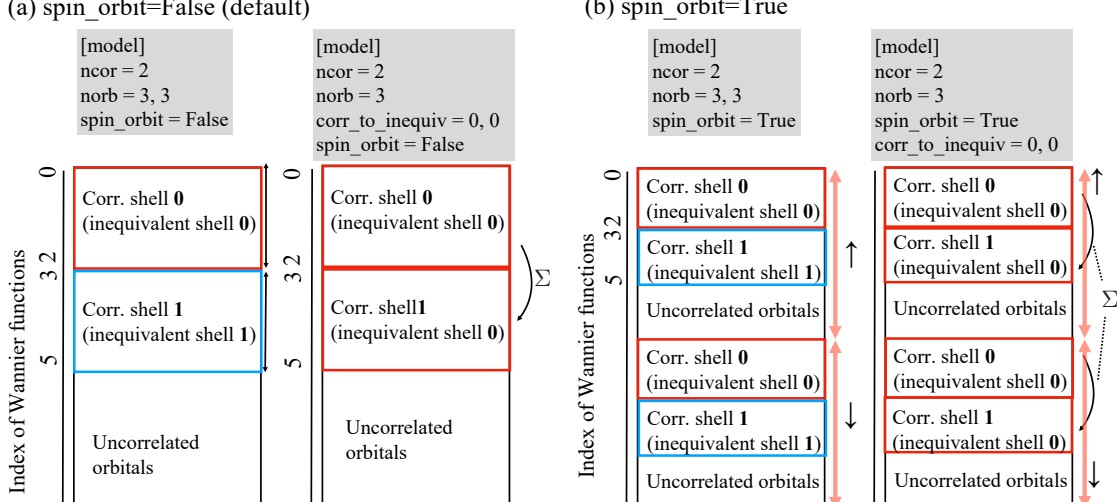

Figure 5: Shell structure of Wannier90 file with (a) spin_orbit=True and (b) spin_orbit=False. As an example, we consider cases with atoms in a unit cell (ncor=2). For the right panels of (a) and (b), the two atoms are treated equivalently (i.e., the self-energy is assumed to be identical).

parameters at each inequivalent shell are specified by the parameter `kanamori` as

```
interaction = kanamori
kanamori = [(U_1, U'_1, J_1), (U_2, U'_2, J_2), ... ].
```

(b) If `interaction = slater_f`, the interaction matrix is constructed by the effective Slater integrals $F_0, F_2, F_4, F_6$ [25]. These Slater integrals and the angular momentum $l$ at each inequivalent shell are specified by the parameter `slater_f` as

```
interaction = slater_f
slater_f = [(angular_momentum, F_0, F_2, F_4, F_6), ... ]
```

It is noted that $F_0, F_2, F_4, F_6$ must all be specified.

(c) If `interaction = slater_uj`, the Slater-type interaction is used. When $U, J$, and the angular momentum $l$ are set, the effective Slater integral is calculated internally in DCore according to the formula defined in Table 2. The $U, J$, and $l$ at each inequivalent shell are specified by the parameter `slater_uj` as

```
interaction = slater_uj
slater_uj = [(angular_momentum1, U1, J1), ... ]
```

Table 2: Formula of the effective Slater integrals for `interaction = slater_uj` [25, 26].

| $l$ | $F_0$ | $F_2$ | $F_4$ | $F_6$ |
|---|---|---|---|---|
| 1 | $U$ | $5J$ | - | - |
| 2 | $U$ | $\frac{14}{1.0+0.63}J$ | $0.63F_2$ | - |
| 3 | $U$ | $\frac{6435}{286+195\times\frac{451}{675}+250\times\frac{1001}{2025}}J$ | $\frac{451}{675}F_2$ | $\frac{1001}{2025}F_2$ |

(iv) Local potential parameters

An arbitrary local potential can be implemented using parameters `local_potential_`

matrix and `local_potential_factor`. `local_potential_matrix` describes, in the Python dictionary format, a set of inequivalent shell indices *ish* and a filename that defines the local potential matrix. The parameter `local_potential_factor` defines a prefactor to the potential matrix. For details, see DCore's online manual.

2. `[system]` block
This block includes thermodynamic parameters and some technical parameters such as the inverse temperature and the number of Matsubara frequencies. It is also possible to specify whether the chemical potential is fixed; if it is fixed, its value can be included. There is also the option `with_dc`, used to consider double-counting correction [see Eq. (15)]. Other parameters are described in the online document [27].

3. `[impurity_solver]` block
This block specifies the impurity solver to be used and the necessary parameters for running the solver program. In addition, the solver-dependent parameters can be specified in this block. For details, see the online document [28].

4. `[control]` block
This block includes parameters that control the self-consistency loop of DMFT. For example, calculation condition parameters, such as the maximum steps of DMFT loops and the convergence criteria, the input file names of the initial self-energies, and some calculation options, such as the restart option, are specified as described in the online document [29].

5. `[tool]` block
This block includes parameters for post calculations of real-frequency quantities, `dcore_post`. A real-frequency mesh and $k$-path are configured here. For more details, see the online document [30].

6. `[mpi]` block
This block includes parameters that are read by `dcore` and `dcore_post`. The usage of the `[mpi]` block is explained in Sec. 6.2 and in the online document [31].

## 4.2 Output-file format

Output files are generated by each program. The output files for each program are described below.

1. `dcore_pre`
`dcore_pre` generates `seedname.h5`, where `seedname` is defined in a `model` block in the input file. It has two groups, namely `dft_input` and `DCore`. See the DFTTools website [13] for details on the data structure in the `dft_input` group. In the `DCore` group, the values of interaction matrix $U_{\alpha\beta\gamma\delta}(s)$ for each equivalent shell , where $\alpha, \beta, \gamma, \delta$ denote the spin-orbital indices at each inequivalent shell, and local potential $V^i_{s,o1,o2}$, where $s$ denotes the spin and $o1, o2$ denote orbitals, are output.

2. `dcore`
`dcore` generates `seedname.out.h5`. All data are stored in the `dmft_out` group. In this group, the input parameters, the total number of iteration steps, and the physical quantities, such as the local self-energy in the imaginary frequency domain and the chemical potential, at each iteration step are output. The data structure of the self-energy is described in the online document [32]. For instance, the self-energy at the first iteration for the 0-th inequivalent shell will be stored below `/dmft_out/Sigma_iw/ite1/sh0`

in `seedname.out.h5`. An external impurity solver is invoked as a subprocess from the main process of `dcore`. For each iteration and each inequivalent shell, `dcore` generates temporary input and output files in the working directory, e.g., `work/imp_shell0_ite1` for the first iteration for the 0-th inequivalent shell. The user can check these temporary files to diagnose problems.

3. `dcore_check`
`dcore_check` generates several pairs of a text file including numerical data and a figure file in PNG format in `check` directory. `sigma.dat` includes the local self-energy $\Sigma^{\text{imp}}(i\omega_n, s)$ at the final step. The corresponding figure file `sigma_ave.png` show the average self-energy of the last seven iterations, where the average is taken by

$$\Sigma_{\text{Ave}}(i\omega_n) = \frac{\sum_{s \in \text{inequivalent shells}} \sum_{\alpha,\beta}^{N_{\text{orb}}^s} \Sigma_{\alpha\beta}(i\omega_n, s)}{\sum_{s \in \text{inequivalent shells}} N_{\text{orb}}^s}. \tag{16}$$

The maximum frequency is specified with the parameter `omega_check` in the `[tool]` block. All other files have a prefix `iter_`, which indicates that the history during the iteration is stored. The text file `iter_mu.dat` contains the chemical potential as a function of the iteration number. `iter_sigma-ish0.dat`, `iter_occup-ish0.dat`, and `iter_spin-ish0.dat` stores the renormalization factor, the occupation number, and the spin moment, respectively. Here, the number in `-ish0` indicates the index for inequivalent shells. All data in the file `iter_*.dat` are plotted in `iter_*.png`.

4. `dcore_post`
`dcore_post` generates three text files, namely `seedname_dos.dat`, which includes the total spectral function, `seedname_akw.dat`, which includes the single-particle excitation spectrum $A(k, w)$, and `seedname_momdist.dat`, which includes the momentum distribution function. A script (`seedname_akw.gp`) for displaying $A(k, w)$ is also generated at the same time. All files and figures are output in the `post` directory.

# 5 Installation

DCore is a collection of pure Python programs depending on TRIQS [11] and TRIQS/DFTTools [13]. DCore supports TRIQS 3.0.x and Python3. Once these prerequisites are installed, the installation of DCore is quick and efficient:

```
$ pip3 install dcore
```

A pure-Python package will be automatically downloaded, being installed into the system site-packages directory. If you do not have root privileges, you can install DCore into the user site-packages directory:

```
$ pip3 install dcore --user
```

You can find the locations of the installed files as

```
$ pip3 show -f dcore
```

Please make sure that the PATH environment variable includes the directory containing the DCore executable files such as dcore, dcore_pre.

The detailed installation instructions can be found online [33]. In addition, at runtime, one of the external impurity solvers shown below must be available.

- ALPS/CT-HYB [15]

- ALPS/CT-HYB-SEGMENT [16]

- TRIQS/cthyb [12]

- TRIQS/Hubbard-I [21]

- pomerol [22] (as Hubbard-I solver and ED solver)

We provide an up-to-date list of supported impurity solvers and short instructions on installation and usage online [34].

# 6 Examples

The examples below show how to define a model, how to compute a self-consistent solution, and how to compute physical quantities.

## 6.1 First example: Square-lattice Hubbard model

As the first example, we consider a single-band Hubbard model on a square lattice. Without the shell index, the Hamiltonian in Eq. (3) is reduced to

$$\mathcal{H} = \sum_{k\sigma} \epsilon_k c_{k\sigma}^\dagger c_{k\sigma} + U \sum_i n_{i\uparrow} n_{i\downarrow}, \tag{17}$$

where $n_{i\sigma} = c_{i\sigma}^\dagger c_{i\sigma}$ is the local number operator. The energy dispersion $\epsilon_k$ is given by

$$\epsilon_k = 2t(\cos k_x + \cos k_y) + 4t' \sin k_x \sin k_y. \tag{18}$$

Note that the nearest-neighbor hopping parameter $t$ should be negative to minimize $\epsilon_k$ at the $\Gamma$ point. This is an effective model of cuprate superconductors and is one of the most studied models in strongly correlated electron systems.

We explain below the steps for obtaining the DMFT solution with the input file `dmft_square_pomerol.ini`, which is available in the online tutorial [35].

### 6.1.1 Model construction

We first set up the model using `dcore_pre`. Parameters for defining a model are provided in the `[model]` block of the input file. The square-lattice Hubbard model can be constructed as shown below.

```
[model]
seedname = square
lattice = square
norb = 1
nelec = 1.0
t = -1.0
kanamori = [(4.0, 0.0, 0.0)]
nk = 8
```

The square lattice is predefined in DCore and can be invoked simply by setting `lattice = square`. For more complicated models that are not predefined in DCore, this parameter is replaced with `lattice = wannier90` and the lattice data are described in a separate file with the Wannier90 format. Sec. 6.1.5 describes its details. We consider half-filling, which is

specified by `nelec = 1.0`. The value of transfer integral $t$ is specified by `t`. In this case, $t$ is set as $-1$, and the absolute value of $t$ is taken as the unit of energy. The interaction parameter $U$ is input by the first component of `kanamori`. The second and third components correspond to $U'$ and $J$, respectively, but are meaningless for single-band models. The system size is determined by `nk = 8`, which means that the $\boldsymbol{k}$-average in Eq. (6) is evaluated by summing up $8 \times 8$ $\boldsymbol{k}$-points.

With this input file, `dcore_pre` is executed as

```
$ dcore_pre dmft_square_pomerol.ini
```

This program generates an HDF5 file named `square.h5`, which includes all the information on the model Hamiltonian, such as $H(k)$.

### 6.1.2 Self-consistent calculations

We now proceed to the main program `dcore`. For executing the program, we need three additional blocks in the parameter file, namely `[system]`, `[impurity_solver]`, and `[control]`. The example below shows a parameter set for a square lattice:

```
[system]
T = 0.1
n_iw = 1000
fix_mu = True
mu = 2.0

[impurity_solver]
name = pomerol
exec_path{str} = pomerol2dcore
n_bath{int} = 3
fit_gtol{float} = 1e-6

[control]
max_step = 100
sigma_mix = 0.5
converge_tol = 1e-5
```

The `[system]` block includes `T` for temperature $T$ and `n_iw` for the number of positive Matsubara frequencies. One can use either the parameter $\beta$ (inverse temperature) or $T$ (temperature) to set the simulation temperature. We fix $\mu$ at 2 by specifying `fix_mu = True` and `mu = 2.0` because the condition for half-filling is known. If `fix_mu` is not activated, $\mu$ is adjusted every time the impurity problem is solved to make the occupation number equal to `nelec`.

The `[impurity_solver]` block specifies which impurity solver is used and its configuration. In this example, we use the ED solver `pomerol` [22], which is specified by `name = pomerol`. Other parameters in this block are solver-dependent and require the type specification, such as `{str}` and `{int}`. `exec_path{str}` is a full/absolute path to the executable (including its name). Here, we assume that the executable `pomerol2dcore` is in the PATH. In ED-based solvers, the hybridization function $\Delta_\sigma^{\alpha\beta}(i\omega_n)$ in Eq. (13) is approximated by the function composed of $N_{\text{bath}}$ poles:

$$\tilde{\Delta}_\sigma^{\alpha\beta}(i\omega_n) = \sum_{b=1}^{N_{\text{bath}}} \frac{V_\sigma^{\alpha b}(V_\sigma^{\beta b})^*}{i\omega_n - E_\sigma^b}. \tag{19}$$

Hereafter, we drop the shell index $s$ for simplicity. The energy level $E_\sigma^b$ and the hybridization parameter $V_\sigma^{\alpha b}$ are determined by fitting $\tilde{\Delta}_\sigma^{\alpha\beta}(i\omega_n)$ to $\Delta_\sigma^{\alpha\beta}(i\omega_n)$ [1]. The fitting tolerance is

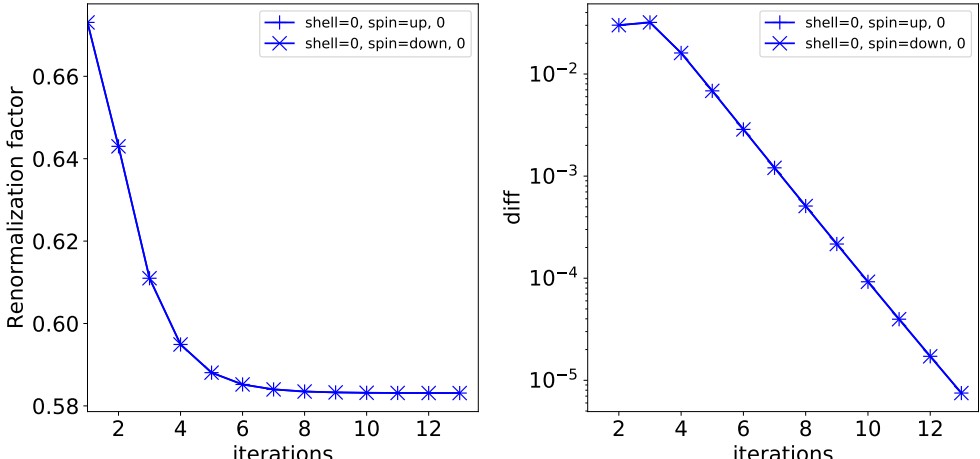

Figure 6: One of the figures generated by `dcore_check` for a convergence check. The renormalization factor $z$ is plotted as a function of the iteration number (Left panel) and its absolute difference between iterations $i$ and $i-1$ is plotted in a logarithmic scale (Right panel).

specified by `fit_gtol`. The resultant finite-size system consisting of the correlated site plus $N_{\text{bath}}$ bath sites is solved using numerical diagonalization.

The [`control`] block includes parameters that control the self-consistent calculations. The impurity problem is solved `max_step = 100` times at maximum, but the loop is terminated if a convergence criterion is satisfied. The criterion we use is

$$|O[i] - O[i-1]| < \texttt{converge\_tol}, \tag{20}$$

where $O[i]$ denotes a quantity at the $i$-th iteration. The quantities $O$ include the chemical potential $\mu$ and the renormalization factor $z$ [see Eq. (21)]. `sigma_mix` specifies the mixing parameter $\sigma_{\text{mix}}$ in Eq. (14). The optimal value of `sigma_mix` depends on models and parameters. Generally, a larger value of `sigma_mix` (1 at the largest) can lead to a faster convergence, but may result in a bad convergence (e.g., oscillation and discontinuous change). A recommended strategy is that one starts from a large value of `sigma_mix` such as 1 and 0.5 and decreases it if the convergence graph (see below) does not show a tendency of convergence.

The main program `dcore` is executed as

```
$ dcore --np 4 dmft_square_pomerol.ini
```

Here, the number of MPI processes that are internally invoked is given after the `-np` option. Note that `dcore` itself is launched as a single process. The results are stored in an HDF5 file named `square.out.h5`.

After `dcore` is executed, one must check the convergence of the self-consistent calculations. For this purpose, the auxiliary program `dcore_check` is provided. `dcore_check` is called without the `-np` option as

```
$ dcore_check dmft_square_pomerol.ini
```

Figure 6 shows one of the figures generated by `dcore_check`. The renormalization factor $z$ defined by

$$z_{\sigma} = [1 - \text{Im}\Sigma_{\sigma}(i\omega_0)/\omega_0]^{-1}, \tag{21}$$

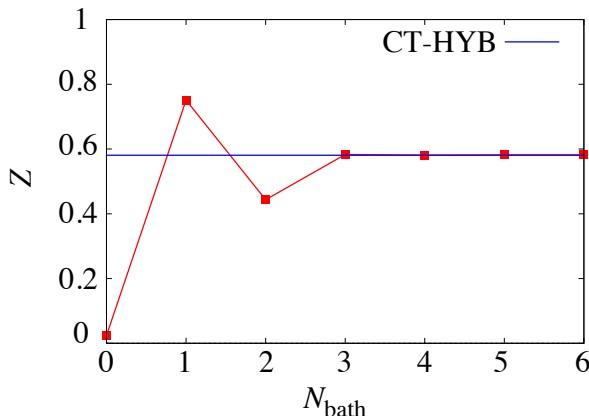

Figure 7: Renormalization factor $z$ as a function of the number of bath sites $N_{bath}$ computed by the `pomerol` solver. The blue line shows the CT-QMC result $z = 0.58$.

is plotted as a function of the iteration number (left panel). $z$ is bounded in the range $0 < z \leq 1$. The right panel shows the difference of the renormalization factor between the $i$-th and $(i-1)$-th iterations. The graph shows an exponential decay. The loop is terminated at the 13th iteration, where the difference falls below `converge_tol = 1e-5`.

If the loop does not converge within `max_step`, one can continue the self-consistent calculations. To this end, one activates the `restart` option by adding a line in the input file:

```
[control]
restart = True
```

Then, `dcore` uses the final result for $\Sigma(i\omega_n)$ in the previous run as the initial state and executes additional `max_step` iterations. We note that the default is `restart=False`, which means the self-consistent loop is started over even though the previous result exists in the current directory. Other parameters in [control] block can be changed in the second run. For example, `sigma_mix` may be reduced if the convergence graph does not show a converging behavior. This procedure is repeated until convergence is reached.

At this point, the DMFT solutions have been obtained for static quantities and $G(\boldsymbol{k}, i\omega_n)$ in the Matsubara domain. For example, the magnetization and the renormalization factor $z$ can be discussed. Figure 7 shows the convergence of $z$ versus the number of bath sites $N_{bath}$. Convergence to the CT-QMC result (explained in Sec. 6.1.4) is observed for $N_{bath} \geq 3$.

### 6.1.3 Dynamical quantities

After the self-consistent calculations are finished, the post calculation `dcore_post` is conducted to compute dynamical quantities on the real frequency axis such as the density of states $A(\omega)$ and the single-particle excitation spectrum $A(\boldsymbol{k}, \omega)$. The parameters for this step are provided in the [tool] block as follows:

```
[tool]
knode = [(G,0,0,0),(X,0.5,0,0),(M,0.5,0.5,0),(G,0,0,0)]
nk_line = 100
omega_max = 6.0
omega_min = -6.0
Nomega = 401
broadening = 0.4
```

The $k$-path is specified by the parameter `knode`. In this example, $k$ starts from Γ and comes back to Γ through X and M. For each interval, the path is divided into `nk_line = 100` points, on which $A(k, \omega)$ is computed. The $\omega$ mesh is generated using the parameters `omega_min`, `omega_max`, and `Nomega`. The parameter `broadening` specifies the extent of an artificial broadening $\delta$ in the spectrum. This is done by replacing $\omega$ by $\omega + i\delta$ in the analytical continuation. `broadening` is set to a value of the order of $T$ to obtain a smooth spectrum with the ED solver. For other solvers that treat the thermodynamic limit, one should set `broadening = 0` to avoid artificial broadening.

dcore_post is executed with the `-np` option because MPI parallel computation is used:

```
$ dcore_post --np 4 dmft_square_pomerol.ini
```

After the computation is finished, numerical results are stored in `post` directory. The spectrum $A(k, \omega)$ on a given $k$-path, for example, is saved in a text file `square_akw.dat` together with a plotting script `square_akw.gp` in gnuplot format. One can plot the data by

```
$ gnuplot square_akw.gp
```

Figure 8(a) shows $A(k, \omega)$ plotted with this command. We note that the range of the colorbar needs to be changed manually to obtain a distinct spectrum. Although the ED solver yielded reasonable estimations for $z$, as shown in Fig. 7, the dynamical quantities clearly show artificial features. Because $\Delta(\omega)$ is composed only of `n_bath = 3` poles located at $\omega = 0$ and $\pm 1.73$, $A(k, \omega)$ exhibits discrete features. However, the hybridization should take place equally in the whole energy region. A larger `n_bath` is necessary to discuss dynamical properties.

We note that the Padé approximation used for the analytical continuation does not ensure the causality, namely, $A(k, \omega)$ may becomes negative at some $(k, \omega)$. One can partly avoid this unphysical behavior by increasing the value of `broadening`. Another risk is involved in the analytical continuation from noisy $\Sigma(i\omega_n)$, which will be described in 6.1.4.

### 6.1.4 CT-QMC solver

The CT-QMC method offers an unbiased simulation of general interacting models [36, 37]. In particular, efficient calculations for the effective impurity problem are achieved by its hybridization-expansion formulation (CT-HYB), which performs Monte Carlo sampling in the expansion with respect to hybridization between the correlated atom and the bath as perturbation [38]. In special cases without exchange interactions, an even more efficient algorithm based on

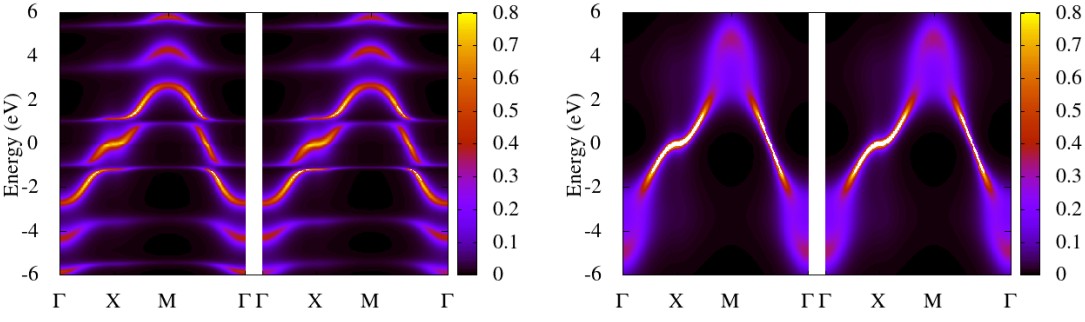

Figure 8: $A(k, \omega)$ for each spin component computed by (a) the ED solver with three bath sites and (b) the CT-QMC solver.

a segment picture can be used [39]. Here, we use ALPS/CT-HYB-SEGMENT, which is the ALPSCore [40] version of the implementation developed by Hafermann *et al.* [41]. A detailed documentation with overview of parameters and sample scripts is provided in the subdirectory `Documentation` in the repository [16].

The segment algorithm can handle only the density-density type interactions. This is the case with the single-orbital Hubbard model and the multi-orbital Hubbard model with $J = 0$. When the segment solver is attempted to use, DCore check if the model includes non-density-density interactions and if yes, the program stops before the impurity solver is invoked. Even in such cases, one can still apply the segment solver by activating the `density_density` option as

```
[model]
density_density = True
```

Then, the interactions generated by `dcore_pre` are restricted to density-density type. Note that the `density_density` option changes the model (e.g. when $J$ is finite) and hence the results may differ from those obtained by other solvers such as ALPS/CT-HYB.

To invoke the CT-HYB solver, the parameters in the `[impurity_solver]` block of the input file are changed as follows:

```
[impurity_solver]
name = ALPS/cthyb-seg
exec_path{str} = /path/to/alps_cthyb
cthyb.TEXT_OUTPUT{int} = 1
MAX_TIME{int} = 60
cthyb.N_MEAS{int} = 50
cthyb.THERMALIZATION{int} = 100000
cthyb.SWEEPS{int} = 100000000
```

The CT-QMC solver is invoked by `name = ALPS/cthyb-seg`. The second parameter `exec_path{str}` should be changed to point to the path to the `alps_cthyb` executable in the environment. Important parameters for the QMC simulation are

- `MAX_TIME{int}`,

- `cthyb.N_MEAS{int}`,

- `cthyb.THERMALIZATION{int}`,

- `cthyb.SWEEPS{int}`.

The simulation finishes either when `cthyb.SWEEPS{int}` Monte Carlo sweeps are finished or when the elapsed time reaches `MAX_TIME` seconds. In this example, `cthyb.SWEEPS{int}` is so large that the computation time is controlled by `MAX_TIME{int}`. `cthyb.N_MEAS{int}` is the number of updates per measurement, and here we use 50 following the example in the official documentation [16]. `cthyb.THERMALIZATION{int}` specifies the number of updates before the measurement starts. `cthyb.THERMALIZATION{int} = 100000` should be large enough to properly discard irrelevant samples. However, this value may not be large enough at lower temperatures. This should be checked by, for example, plotting the histogram of the expansion order [41]. The whole list of optional parameters can be found by running the command

```
$ alps_cthyb --help
```

Note that solver-dependent parameters should be input with the type specification, e.g., `{int}` and `{str}`, as in the above example.

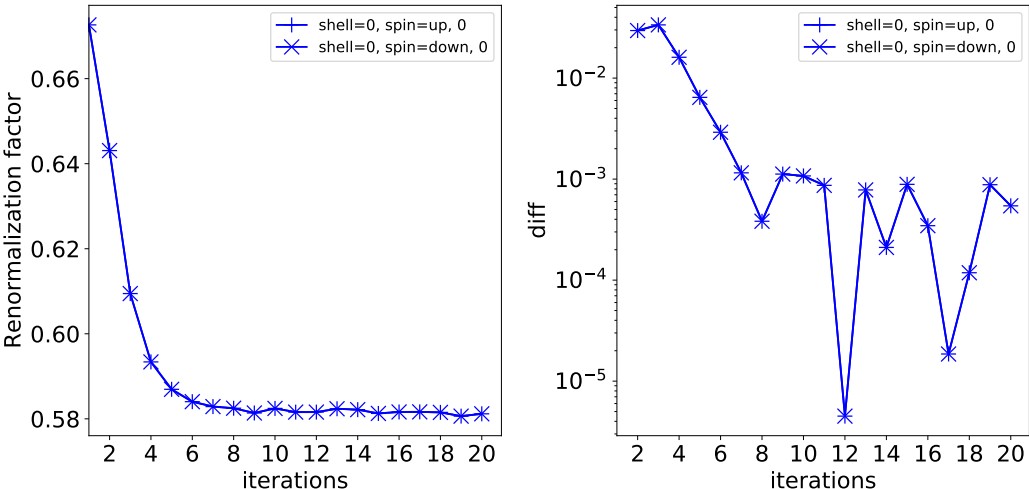

Figure 9: Case of CT-HYB solver for a figure generated by `dcore_check` for a convergence check. See the caption in Fig. 6.

Because the QMC solver is time-consuming, it is important to use a proper initial guess for $\Sigma(i\omega_n)$ to reduce the number of iterations. For example, we can use the converged result obtained for similar parameters or the result obtained for the same parameters with a different solver. This can be done by assigning the path of `sigma.dat` generated by `dcore_check` to the `initial_self_energy` parameter as

```
[control]
initial_self_energy = /path/to/sigma.dat
max_step = 20
sigma_mix = 0.5
time_reversal = True
```

Here, /path/to/sigma.dat should be changed according to the environment. The parameter `time_reversal = True` activates the spin average of $\Sigma_\sigma(i\omega_n)$. This is particularly useful for the QMC solver to improve statistics. Unlike the previous example using the ED solver, `converge_tol` is not specified here, because statistical errors in the QMC method make the automatic convergence check less reliable. The convergence is checked visually using the graphs generated by `dcore_check`.

With this input file, all programs, namely `dcore_pre`, `dcore`, `dcore_check`, and `dcore_post`, are executed as in Sec. 6.1.1. Figure 9 shows graphs generated by `dcore_check`. The right panel shows that the statistical error is on the order of $10^{-3}$ and that convergence is reached at around the 10th iteration. The result for $A(\mathbf{k}, \omega)$ is shown in Fig. 8(b). Note that artificial broadening is turned off by `broadening = 0.0`. The spectrum exhibits low-energy quasiparticle excitations and high-energy broadening due to correlations.

We note that spectra computed using the Padé approximation is extremely sensitive to statistical errors. For this reason, Fig. 8(b) might not be reproduced even with the same input as above. In such cases, one can try improving the QMC accuracy by increasing the number of MPI processes or by increasing `MAX_TIME{int}` parameter. An alternative way for analytical continuation is discussed in Sec. 7 as a future development.

Here, a comment on the convergence check is in order. Although the automatic convergence check has been turned off in the above example, there is a way to employ the convergence check even for QMC solvers. In the case with Fig. 9, the following configuration works:

```
[control]
converge_tol = 0.002
n_converge = 5
```

The DMFT loop is terminated when the convergence criterion (20) is satisfied `n_converge` times consecutively. `converge_tol` is set larger than the magnitude of statistical errors, and the other parameter `n_converge=5` prevents the loop from being terminated prematurely. This way of automatic convergence check can be attempted to use if the expected magnitude of statistical errors is known in advance.

As discussed above, one could alternatively use a CT-HYB solver based on the matrix algorithm, such as ALPS/CT-HYB [see Sec. 6.2]. To obtain the same results with comparable accuracy, one may have to run a CT-HYB solver based on the matrix algorithm longer in runtime, typically by one order of magnitude.

### 6.1.5 Wannier90 interface

DCore provides simple lattice models such as a one-dimensional chain, a two-dimensional square lattice, and a three-dimensional cubic lattice. Arbitrary lattices that are not predefined in DCore can be implemented using the Wannier90 format [42]. In the following, we show how to construct the square lattice using Wannier90 input.

The only difference in `dmft_square_pomerol.ini` is in the [model] block, which includes

```
[model]
seedname = square
lattice = wannier90
norb = 1
nelec = 1.0
kanamori = [(4.0, 0.0, 0.0)]
nk = 8
```

The difference between this block and that in Sec. 6.1.1 is `lattice = wannier90`. When this is given, DCore reads a wannier90 file named *seedname*_hr.dat (square_hr.dat in the present case) to set up the one-body part of the Hamiltonian, $H(\boldsymbol{k})$. square_hr.dat describing the square lattice is given by

```
square lattice
1
9
1 1 1 1 1 1 1 1 1
 0  0 0 1 1   0.0 0.0
 0  1 0 1 1  -1.0 0.0
 1  0 0 1 1  -1.0 0.0
 0 -1 0 1 1  -1.0 0.0
-1  0 0 1 1  -1.0 0.0
 1  1 0 1 1   0.0 0.0
 1 -1 0 1 1   0.0 0.0
-1  1 0 1 1   0.0 0.0
-1 -1 0 1 1   0.0 0.0
```

The first line can be any text. The second line specifies the number of orbitals in a unit cell. The next line specifies the number of cells, $N_{\text{cell}}$, for which the transfer integrals are provided. Here, we have $N_{\text{cell}} = 9$ cells, as shown in Fig. 10(a). The value of transfer integrals, $H_{ij}^{\text{W90}}(\boldsymbol{R})$, in the one-body Hamiltonian $\mathcal{H} = \sum_{\boldsymbol{R}\boldsymbol{R}'ij} H_{ij}^{\text{W90}}(\boldsymbol{R}' - \boldsymbol{R})c_{\boldsymbol{R}i}^{\dagger}c_{\boldsymbol{R}'j}$ is listed for all possible combinations

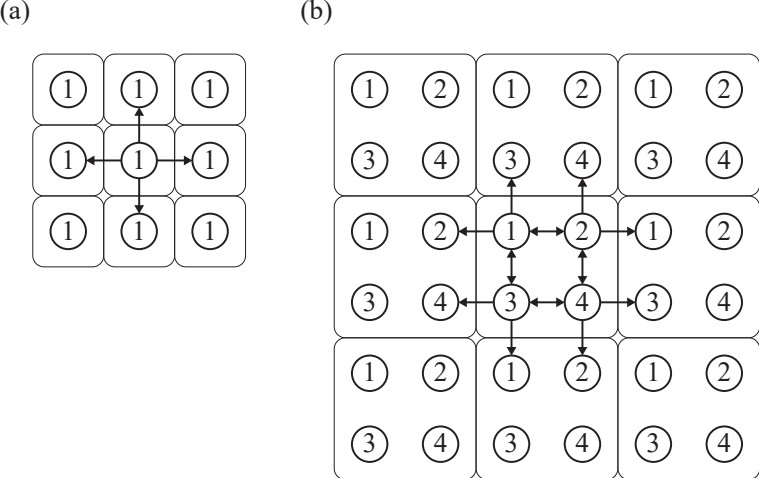

Figure 10: Lattice configuration for the Wannier90 input for (a) the paramagnetic state and (b) the AFM state. The circles show sites (atoms) and the number is the label of the site in the unit cell indicated by the square. The arrows indicate the hopping included in the input file.

of $\boldsymbol{R}$, $i$, and $j$ for fixed $\boldsymbol{R}' = 0$. Here, $i$ and $j$ denote the combined orbital and site indices in the unit cell. The format is "$l_1\ l_2\ l_3\ i\ j\ \mathrm{Re}\,H_{ij}^{\mathrm{W90}}(\boldsymbol{R})\ \mathrm{Im}\,H_{ij}^{\mathrm{W90}}(\boldsymbol{R})$", where $\boldsymbol{R} \equiv l_1\boldsymbol{a}_1 + l_2\boldsymbol{a}_2 + l_3\boldsymbol{a}_3$ and $(\boldsymbol{a}_1, \boldsymbol{a}_2, \boldsymbol{a}_3)$ are the unit lattice vectors. The convention used by Wannier90 is related to the one in Eq. (4) as $H_{ij}^{\mathrm{W90}}(-\boldsymbol{R}) = H_{ij}(\boldsymbol{R})$.

### 6.1.6 Antiferromagnetic state

With the Wannier90 format, one can construct arbitrary lattices. For example, we can enlarge the unit cell to allow an antiferromagnetic (AFM) solution. Let us see how this is done for the square lattice example.

Figure 10(b) shows the lattice configuration for AFM calculations. Four "orbitals" are contained in a unit cell. The Wannier90 format file now includes the hopping inside the unit cell (8 double-pointed arrows) and the hopping from the central cell to neighboring unit cells (8 single-pointed arrows). The full input files as well as a script for generating the Wannier90 format file are available in the online tutorial [43].

The parameters in the [model] block of the input file are given as follow:

```
[model]
seedname = afm_dim2
lattice = wannier90
ncor = 4
norb = 1, 1
nelec = 4.0
kanamori = [(4.0, 0.0, 0.0), (4.0, 0.0, 0.0)]
corr_to_inequiv = 0, 1, 1, 0
nk = 8
```

The total number of orbitals contained in the unit cell is specified by `ncor = 4`. `nelec` is the total number of electrons in the unit cell, and hence needs to be multiplied by 4. In the AFM state, we expect a staggered ordered state, where site 1 and site 4 are equivalent (e.g., spin-up state), and site 2 and site 3 are equivalent (spin-down state). This constraint can be imposed by setting `corr_to_inequiv = 0, 1, 1, 0`, which indicates that there are only

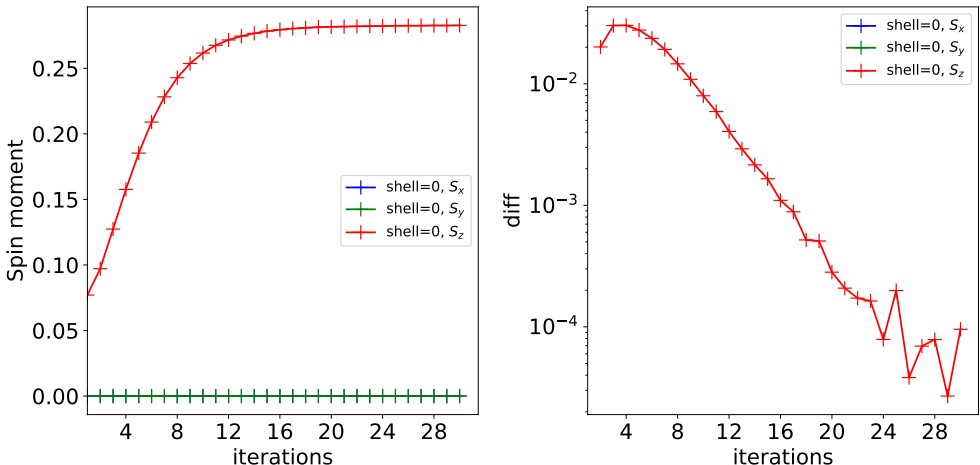

Figure 11: Spin moment $m_\xi(s)$ ($\xi = x, y, z$) for inequivalent site $s = 0$ as a function of the iteration number. The left panel shows $m_\xi(s)$ itself and the right panel shows the difference between iterations $i$ and $i - 1$.

two inequivalent sites: site 1 and site 4 are labeled with 0, and site 2 and site 3 are labeled with 1. Two independent impurity problems are solved in each step of the self-consistent calculations. For each impurity model, the number of orbitals and the strength of interactions are specified by the `norb` and `kanamori` parameters, respectively.

To obtain an AFM solution, we need to start the self-consistent calculation with a broken-symmetry initial guess. To this end, we set the initial self-energy to

$$\Sigma_{\alpha\beta}^\sigma(i\omega_n, s) = v_{\alpha\beta}^\sigma(s), \tag{22}$$

using local potential $v_{\alpha\beta}^\sigma(s)$, which depends on the inequivalent site $s$. The [control] block contains an additional parameter that specifies the initial self-energy as

```
[control]
max_step = 30
sigma_mix = 0.5
initial_static_self_energy = {0: 'init_se_up.txt', 1: 'init_se_down.txt'}
```

A path to a text file is given in `initial_static_self_energy` for each inequivalent site using the dictionary literal in Python. The text files describe non-zero components in $v_{\alpha\beta}^\sigma$ in the form "$\sigma$ $\alpha$ $\beta$ Re $v_{\alpha\beta}^\sigma$ Im $v_{\alpha\beta}^\sigma$". To obtain a magnetic state (spin-up state), we can input the magnetic field in `init_se_up.txt` as

```
0 0 0 -1.0 0.0
1 0 0 1.0 0.0
```

and the magnetic field with the opposite direction is given in `init_se_down.txt` as

```
0 0 0 1.0 0.0
1 0 0 -1.0 0.0
```

The parameters in [system] and [impurity_solver] do not change from the paramagnetic calculations.

Figure 11 shows the convergence of the spin moment $m_\xi(s)$ ($\xi = x, y, z$) for inequivalent site $s = 0$. A convergence of $m_z$ to a non-zero value was obtained around the 30th iteration. The moment $m_z = 0.283$ is about half the full moment (1/2). We confirmed that $m_z(s)$ for

$s = 1$ converges to the negative side, namely $m_z(0) = -m_z(1)$ (no figure). The parameters $U = 4$, $n = 1$, and $T = 0.1$ are the same as those in the previous calculations in Fig. 8, meaning that the paramagnetic solution was not a true solution of the DMFT equation. The phase diagram is shown, for example, in Ref. [44].

## 6.2 Solving multi-orbital model using QMC solver: $t_{2g}$ model on a Bethe lattice

In this subsection, we show how to solve a three-orbital model on a Bethe lattice by using DCore. In this model, an interesting phenomenon called spin-freezing transition occurs [45]. Spin-freezing is signaled by a peculiar frequency dependence of the self-energy: $\text{Im}\Sigma(i\omega_n) \propto \omega_n^{0.5}$. We will reproduce this result using DCore.

### 6.2.1 Model definition

We first make the input file of dcore_pre for generating an HDF5 file that is necessary for DMFT calculations. The Hamiltonian in Ref. [45] is defined as

$$\mathcal{H} = \sum_{k,\sigma} \epsilon_k^\alpha c_{k\alpha\sigma}^\dagger c_{k\alpha\sigma} + \sum_{\boldsymbol{R}} \mathcal{H}_{\text{loc}}(\boldsymbol{R}), \tag{23}$$

$$\mathcal{H}_{\text{loc}}(\boldsymbol{R}) = -\sum_{\alpha,\sigma} \mu n_{\boldsymbol{R}\alpha\sigma} + \sum_\alpha U n_{\boldsymbol{R}\alpha\uparrow} n_{\boldsymbol{R}\alpha\downarrow} + \sum_{\alpha>\beta,\sigma} U' n_{\boldsymbol{R}\alpha\sigma} n_{\boldsymbol{R}\beta-\sigma} + (U'-J) n_{\boldsymbol{R}\alpha\sigma} n_{\boldsymbol{R}\beta\sigma}$$
$$- \sum_{\alpha\neq\beta} J(c_{\boldsymbol{R}\alpha\downarrow}^\dagger c_{\boldsymbol{R}\beta\uparrow}^\dagger c_{\boldsymbol{R}\beta\downarrow} c_{\boldsymbol{R}\alpha\uparrow} + c_{\boldsymbol{R}\beta\uparrow}^\dagger c_{\boldsymbol{R}\beta\downarrow}^\dagger c_{\boldsymbol{R}\alpha\uparrow} c_{\boldsymbol{R}\alpha\downarrow} + \text{H.c.}), \tag{24}$$

where $\alpha = 1, 2, 3$ is the orbital index. We use a Kanamori interaction with $U = 8$, $U' = U-2J = 5.3333333$, and $J = 1.33333$. We use the same model parameters as those used in the paper [45]: $n = 1.6, t = 1.0, U/t = 8.0, J/U = 1/6$. The input file for dcore_pre is as follows:

```
[model]
lattice = bethe
seedname = bethe
nelec = 1.6
t=1.0
norb = 3
kanamori = [(8.0, 5.3333333, 1.33333)]
nk = 1000
```

After running dcore_pre, an HDF5 file named bethe.h5 is generated. DCore discretizes the density of states of the Bethe lattice, a semicircular density of states on $[-2t, 2t]$, along a virtual one-dimensional $k$ axis with nk = 1000 points.

### 6.2.2 DMFT calculation

Next, we make the input file of dcore. The calculation in Ref. [45] was done using the matrix formalism of the CT-HYB method. To make a direct comparison, we use an implementation of the same algorithm, ALPS/CT-HYB, which was developed by one of the authors. An example input file of dcore is shown below.

```
[mpi]
command = '$MPIRUN -np #'

[system]
beta = 50.0

[impurity_solver]
```

```
name = ALPS/cthyb
timelimit{int} = 300
exec_path{str} = hybmat

[control]
max_step = 40
sigma_mix = 1.0
restart = False
```

In this example file, we omit the `model` block, which is the same as that in the input file of
`dcore_pre`. The inverse temperature $\beta$ is set to $50t$. We also define a command for MPI
parallelization using the `mpi` block. Here, `#` in `command` of the mpi block is replaced by the
number of processes specified at runtime. In the above example, we define the environment
variable MPIRUN and run the program with 24 MPI processes as follows.

```
$ export MPIRUN="mpirun"
$ dcore dmft_bethe.ini --np 24
```

After running `dcore`, the results of the self-energy and Green's functions in each iteration are
accumulated in an HDF5 file named *seedname*.out.h5 (`bethe.out.h5` in the present
case) and a text file that contains the self-energies (`sigma.dat`) is output in the `check` direc-
tory.

Before analyzing the computed results such as the self-energy, the user may check the
convergence of CT-QMC sampling. The impurity solver runs for 300 seconds at each iteration
step, and the standard output and error of the solver are redirected to the standard output
of `dcore`. The last part of the output of the solver at the last iteration looks as follows. The
perturbation orders measured just before and after the measurement steps are close to each
other ($\simeq 58$), indicating that the thermalization time was long enough: ALPS/CT-HYB used 10
% of the total simulation time (`timelimit`) for thermalization. One can see that the average
sign is close to 1, indicating that there is no severe sign problem.

```
==== Thermalization analysis ====
Perturbation orders just before and after measurement steps are 57.6833
and 57.9479.

==== Number of Monte Carlo steps spent in configuration spaces ====
Z function: 5.03099
G1: 4.97089

==== Acceptance updates of operators hybridized with bath ====
 1-pair_insertion_remover: 0.177554
 2-pair_insertion_remover: 0.0142077
 Single_operator_shift_updater: 0.196652
 Operator_pair_flavor_updater: 0.166643

==== Acceptance rates of worm updates ====
 G1_ins_rem: 0.0339203
 G1_mover: 0.214957
 G1_flavor_changer: 0.0104972
 G1_ins_rem_hyb: 0.139718
Average sign is  0.999800707488.

Total charge of impurity problem: 1.611984
```

After checking the convergence of the self-consistent iterations using `dcore_check`, one can plot the self-energy stored in `sigma.dat` as shown in Fig. 12. The solid line shows the results taken from [45]. In the low-frequency region, $\text{Im}\Sigma(i\omega_n)$ is proportional to $\omega_n^{1/2}$. The three symbols represent the orbital diagonal components of $\text{Im}\Sigma(i\omega_n)$ computed by DCore. The results obtained by `dcore` match those of [45].

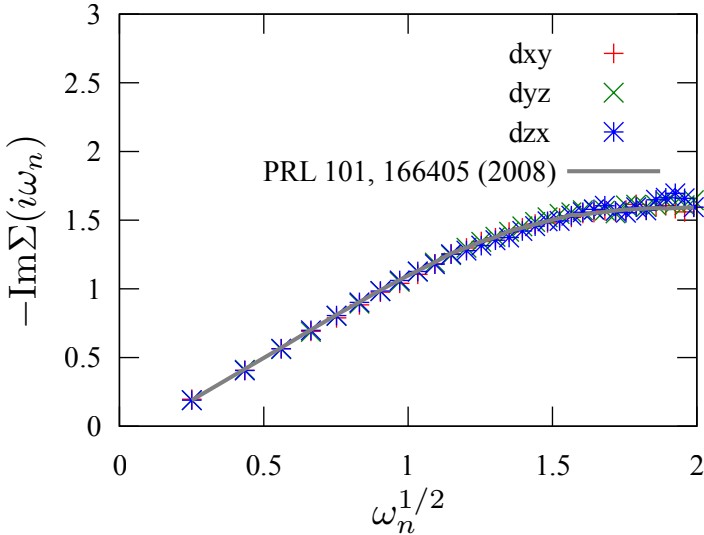

Figure 12: Matsubara frequency dependence of $\text{Im}\Sigma(i\omega_n)$. The solid line indicates the numerical results reported in Ref. [45]. The three symbols indicate $\text{Im}\Sigma(i\omega_n)$ of $d_{xy}$, $d_{yz}$, and $d_{zx}$ orbitals, respectively, obtained by DCore.

## 6.3 DFT+DMFT example: SrVO$_3$

In this section, we demonstrate how to perform a multi-orbital DFT+DMFT calculation for a realistic band structure. In particular, we use SrVO$_3$ which has been studied by DMFT as a testbed in many previous studies.

### 6.3.1 Construction of Wannier functions

We fist construct maximally localized Wannier functions for the $t_{2g}$ manifold using Quantum ESPRESSO [46] and Wannier90 [42]. Any DFT programs that support Wannier90 may be used alternatively. For our DFT calculation, we take a lattice constant of 7.29738 a.u and construct a three-orbital tight-binding model following a common procedure using the local density approximation (LDA). We provide a more detailed description on how to construct Wannier functions using Quantum ESPRESSO or another DFT program, OpenMX [47] online [48]. There, we can download input files as well as the data file of the resultant tight-binding model.

### 6.3.2 Model definition

We show the `model` block of our input file for performing a DMFT calculation using the Wannier functions below. We adopted the Kanamori interaction parameters ($U = 3.44$ eV, $U' = 2.49$ eV, $J = 0.46$ eV) estimated in Ref. [49]. The number of grid points in the first Brillouin zone along each reciprocal vector is set to 10. This number does not have to match the one used in the DFT calculation. The total number of $k$ points is $10^3$.

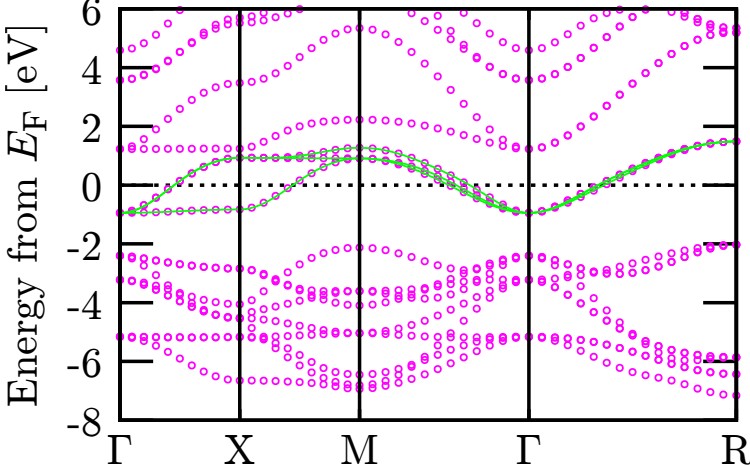

Figure 13: Comparison of LDA band structure and fit by the Wannier functions. The circles denote the energies computed by LDA and the solid lines denote the Wannier fit.

```
[model]
lattice = wannier90
seedname = srvo3
nelec = 1.0
ncor = 1
norb = 3
kanamori = [(3.44, 2.49, 0.46)]
bvec=[(1.627091,0.0,0.0),(0.0,1.627091,0.0),(0.0,0.0,1.627091)]
nk = 10
```

### 6.3.3 Parameters for self-consistent calculations and post processing

We set input parameters for ALPS/CT-HYB in the `impurity_solver` block. The parameters `timelimit` and `exec_path` are passed to ALPS/CT-HYB. The impurity solver runs for 100 seconds with 96 MPI processes at each self-consistent iteration. We set an initial guess for the chemical potential to an appropriate value and inverse temperature to $\beta = 20$ eV$^{-1} \simeq 580$ K. In the input file shown below, we specify the temperature by using the parameter `beta` instead of the parameter T for the sake of brevity in appearance.

```
[mpi]
command = '$MPIRUN -np #'

[system]
beta = 20.0
mu = 12.290722

[impurity_solver]
name = ALPS/cthyb
timelimit{int} = 400
exec_path{str} = hybmat

[control]
max_step = 20
time_reversal = True
```

```
sigma_mix = 0.8
```

The `tool` block includes input parameters for post processing. In this example, we compute $A(k, \omega)$ using `dcore_post` along the $k$-paths specified by symmetry points given in `knode`. As set by the parameter `omega_pade`, the data of the self-energy in the energy window of $0 < \omega_n \leq 2$ is used for analytic continuation to the real-frequency axis. Here, we introduce a small broadening factor for computing spectral functions.

```
[tool]
broadening = 0.01
nk_line = 50
knode=[(G,0,0,0),(X,0.5,0,0),(M,0.5,0.5,0),(G,0,0,0),(R,0.5,0.5,0.5)]
omega_max = 2.0
omega_min = -2.0
Nomega = 400
omega_check = 30.0
omega_pade = 2.0
```

### 6.3.4 Results

In Fig. 13, one can see that the $t_{2g}$ bands are fitted very well by the Wannier functions. The $t_{2g}$ manifold is separated in energy space from the $e_g$ bands above 1.5 eV and oxygen bands below $-2$ eV. Since we constructed the $t_{2g}$ model, the $e_g$ bands and oxygen bands are not taken into account in the present DMFT calculation. Figure 14 shows $A(\boldsymbol{k}, \omega)$ computed by DMFT Compared to the DFT band structure in Fig. 13, one can see that the $t_{2g}$ band is substantially renormalized by strong correlation effects as expected.

There are several previous DMFT studies for this compounds [50–52]. These calculations aimed at a quantitative description of the spectrum of the real material and therefore took into account other bands such as the $e_g$ bands and oxygen bands. Thus, a direct comparison with our result is not possible.

For DCore, one could take into account those *uncorrelated* bands by constructing Wannier functions for the full $3d$ orbitals and oxygen $p$ orbitals in a wider energy window. In this case, we may have to carefully adjust (manually) the level splitting between $t_{2g}$ and the uncorrelated orbitals because the current version of DCore does not support intra-shell (site) interactions between the $d$ and $p$ shells.

We need extra care for quantum Monte Carlo simulations since they often suffer from a sign problem and many other technical issues. If the reader faces such a problem, it is recommended to create an issue on GitHub at the issues page [53] or directly contact the developers of the solver.

## 7   Conclusion

We introduced the open-source software package DCore, which is integrated DMFT software for correlated electrons. We described the algorithm, the structure of the package, installation, and file format, and demonstrated usage for the single-/multi-orbital Hubbard models and the Wannier90 model for $SrVO_3$.

We are planning to implement more functionality in a future version. The current version supports only the Padé approximation for analytic continuation. However, the Padé approximation is sensitive to statistical noise. A future version will support external analytic continuation solvers based on more stable methods such as `SpM` [54] and `Maxent` [55]. Furthermore, we will implement recently proposed efficient methods for computing static/dynamical lattice

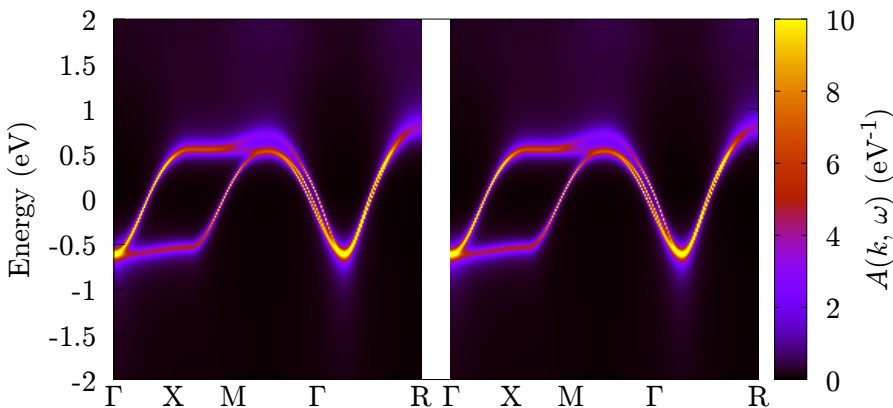

Figure 14: $A(\boldsymbol{k}, \omega)$ for each spin component computed by DMFT for the Wannier model of $SrVO_3$.

susceptibilities based on the Bethe-Salpeter equation [44,56]. There are also plans to support more external impurity solvers such as iQIST [57] and w2dynamics [18].

# Acknowledgements

We acknowledge Takeo Kato and Yuichi Motoyama for supporting the DCore project. We acknowledge the TRIQS community and the ALPS community for developing useful open-source libraries.

**Author contributions** For an updated record of individual contributions, consult the repository at https://issp-center-dev.github.io/DCore/master/index.html.

**Funding information** H.S., J.O., and K.Y. were supported by JSPS KAKENHI Grant No. 18H01158, 21H01003, 21H01041. H.S. was supported by JSPS KAKENHI Grant No. 16K17735. J.O. was supported by JSPS KAKENHI Grant No. 18H04301 (J-Physics). N.T. was supported by JSPS KAKENHI Grant No. 19H05817, and 19H05820. K.Y. was supported by JSPS KAKENHI Grant No. 19K03649, and Building of Consortia for the Development of Human Resources in Science and Technology, MEXT, Japan.

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
