# Peer review of "DCore: Integrated DMFT software for correlated electrons"

_SciPost Physics, doi:SciPost Phys. 10, 117 (2021)_

## Round 1 · Referee Report · Anonymous (Referee 1) · 2020-8-6

Strengths

1- Flexible and easy to use package for DMFT calculations supporting state-of-the-art impurity solvers and ab-initio models constructed from DFT

2- Structure and quality of the presentation in the paper very good in general

3- Nice examples for typical kinds of calculations

Weaknesses

1- Several minor issues with clarity of the presentation

2- Explanations in parts not quite complete enough in my opinion

Report

The authors present a new program package for performing DMFT calculations for multi-orbital Hubbard models. Supported are both simple predefined lattice models with user-specified parameters as well as ab-initio models constructed from DFT using a Hamiltonian in Wannier90 format. Interfaces to multiple existing solvers for the auxiliary Anderson impurity model are implemented and the user can flexibly choose which one to use, including multiple state-of-the-art CT-QMC solvers from the ALPS and TRIQS projects and an exact diagonalization solver.

While there are a few packages for such calculations already and writing a simple DMFT implementation is not very complex, an additional package, especially if it is applicable to a wide range of models, featureful, easy to use, and open-source (and even interfaces with multiple different impurity solvers) is certainly useful and if the package actually ends up helping people do DMFT calculations without the need for 'extensive expert knowledge', as stated in the introduction, a welcome addition.

The presentation in the paper is well structured in my opinion, giving first the background, then an overview of the structure of the package, detailing the input format for the specification of the model and configuration of the calculation and the output format of the results before coming to the technical detail of the installation procedure and presenting a few helpful examples illustrating how input is provided and results evaluated for some typical models.

All in all, I think that both the presented program package as well as this paper are very nice work and that it is suitable for publication with at most minor revisions necessary. However, I have also noticed that a new journal 'SciPost Physics Codebases' apparently has been or is about to be launched, which would usually be a better fit for a paper like this one, but given that I saw no trace of that yet when this paper was first submitted and similar articles have been published in this journal before, that might not be relevant at the moment.

In the following paragraphs, I give short comments on each section individually pointing out some minor issues I found that need revision or could be improved in my opinion.

The introduction is good, although it might be helpful to additionally add references to the applications of DFT+DMFT listed at the end of the first paragraph, and in their mention of the TRIQS project, the authors almost certainly meant to write 'implement their own DMFT code' rather than 'DFT code'.

Section 2 on the background is rather short, but given that the program does not rely on a specific type of impurity solver and the DMFT algorithm can in principle be summarized very succinctly I do not think that this is a problem, although a more detailed description could be added in some places, for example on the adjustment of mu which is just mentioned in passing or the self-energy symmetrization and double-counting correction mentioned in 2.4. Most importantly, some minor issues should be corrected to improve the clarity of the presentation. In particular, some issues I noticed are - in the first term of eq. (1) i and j go over the elements of 'crsh', while the following text explains explicitly that this sum goes over all shells - it is not explained how orbitals are or should be grouped into shells - the text below (6) and in the caption of Fig. 1 say that the k-average is denoted by <...>, while actually <...>_k is used - the self-energy sometimes does and sometimes does not have an extra S parameter that 'indexes correated shells' with no extra explanation, which might surprise readers unfamiliar with the subject - the hybridization function being a matrix is mentioned before it is ever introduced, possibly where the self-energy should have been mentioned instead - Fig. 1 is never explicitly referred to in the text Also, the only reference in the entire section is to a review suggested as further reading, which differs a little in notation from the presentation here and is so comprehensive that the authors might help readers who wish to quickly read up on details by providing more specific guidance or additional references to material that is shorter or has a more introductory-level approach. Further, it is claimed that several methods of double-counting correction are supported, but later one specific formula is given as the used correction in eq. (8) (which should also carry an explicit reference to the appropriate literature) and I was not able to find any other indications of multiple supported methods either here or in the parts of the online manual I read. It is not clear to me what specifically distinguishes the points listed in section 2.4 from other functionality either. The purpose of that section should be stated more clearly (or possibly the content reorganized) and adding at least slightly more details to each point there could help clarifying the intended use cases and specific algorithmic / implementation details (with explicit literature references if appropriate).

Section 3 nicely lays out the structure of the package. The only minor issues I can possibly think of are that the figure suggests two different input files for dcore_pre and dcore while the text specifies a single text file, that one single arrow pointing from impurity solvers to dcore is probably not the best way to graphically represent that relationship (it is at least not the same as in the other cases where such an arrow is used, perhaps a double-headed arrow or two arrows would be better), and that the dcore_check program is missing from this section entirely for some reason.

Section 4.1 is a good summary of the available input parameters, but only the parameters of the model block are explained in detail. I think that explanations how the important parameters of the other blocks, especially system, should be used, possibly even just their names which are mostly missing now, would be helpful additions; if more detailed explanations are not given, at least a more explicit reference to the online reference manual should be added. Apart from that, minor issues are that the 'included blocks' sentence in the first paragraph, apart from the typo, could be expressed more clearly (i.e. that all programs usually read the same file, but that some blocks are, even if present, irrelevant to some of the programs) and that the image for the 'cubic' lattice seems a little visually cluttered and possibly confusing to me, partially also due to the inconsistency that the red cube does not indicate a unit cell while the red squares in the 'square' case do if I interpret the images correctly.

Section 4.2 is a good summary as well. Points that I would clarify are what the 'kind' of correlated shell specifically refers to in the paragraph on dcore_pre and what quantity specifically is called the 'density of states' in this context in the paragraph on dcore_post. There are some potentially confusing typos here as well: the dcore paragraph incorrectly begins with 'dcore_pre' and the parenthesized 'renormalization factor' is placed between 'chemical' and 'potential' instead of after both above eq. (9).

The installation section currently only refers to the online documentation. I think it would be better to give the installation commands at least for the simplest case (e.g. assuming some reasonably normal configuration, that the prerequisites are already installed and that no problems occur) in the paper as well, e.g. four lines for cloning the repository, calling cmake, make and make install or something short like that, so that a reader could follow the paper from beginning to end without having to consult much (or ideally any) additional material under ideal conditions.

Section 6 presents three good examples (the first with multiple slightly different calculations) that nicely illustrate typical use cases: First one very simple square-lattice one-orbital model treated both with exact diagonalization and CT-QMC and then used to demonstrate the Wannier90 interface and calculations with antiferromagnetism, then a multi-orbital model with an interesting physical feature from recent research, and finally a multi-orbital model for the t_2g manifold of SrVO_3 using a Wannier Hamiltonian from DFT as input data.

Minor issues in section 6.1.2 are the in my opinion unclear explanation of the averaging influenced by sigma_mix (i.e. what exactly is averaged and when) and lack of an explanation of what the particular significance of its numerical value is, the slightly ambiguous phrasing of what exactly is executed in parallel at the bottom of the same page, and the description of the restart option, in particular what exactly it does, how it differs from just increasing max_step if it does, and, in that case, why it might be preferable to use it. Further, I find the advice to reduce sigma_mix if the convergence graph exhibits oscillations surprising; I would rather have expected the opposite here if anything, assuming that a higher numerical value of sigma_mix means more influence from earlier self-energy, but then again sigma_mix is not really explained in detail as mentioned. Also, considering that use by non-experts is a stated goal, I am not sure if it is very helpful to instruct the user to just enter some values of unclear origin like 0.5 or 1.0 for not well explained parameters like sigma_mix; I do not think that, having read this paper, a non-expert could be confident in the choice of parameters like sigma_mix. Advice like to change sigma_mix in case of oscillations is probably helpful in such cases (if correct), but advice on how to set it to begin with or specifying whether it is usually even necessary to set it manually would be a good addition. To the parameters of the CT-HYB-SEGMENT solver in 6.1.4 something similar applies, but if that solver is a special case (as the ALPS CT-HYB solver is later used with only a timelimit as configuration) then it would be good to point out explicitly that different solvers might require different amounts of configuration to get good results (and maybe also which solver should be used to avoid manual configuration as much as possible).

In section 6.1.3, the analytical continuation is glossed over, and it is not explained in depth anywhere else in the paper either. Considering the potential influence on the quality of the results, perhaps at least some short advice or a reference in case of problems in the users' own calculations might be appropriate. Further, in my own attempt to reproduce the results shown here, A(k, omega) looked significantly different from Fig. 7a. The artificial features that are explained to come from the discretization to three bath sites were not recognizable, instead even with n_bath = 3 and using the ED solver my results looked more like 7b than 7a. Maybe the authors can think of a reason for this, possibly I made a mistake somewhere, in that case the relevant step should probably be explained more clearly and otherwise the wrong part be fixed. For the plots similar to Fig. 7 it might be also helpful to note that to generate as nice plots as the authors show, the range of the colorbar might need to be restricted manually. The plot scripts generated by dcore_post do not restrict the range, which can result in a rather homogeneous look due to tiny outlier spots, as it happens in Fig. 13 for example.

The only minor issues with 6.1.4 are, as mentioned, that it might be hard for non-experts to know what values to choose for some of the CT-QMC solver parameters (N_MEAS is for example mentioned as an important parameter but not explained at all) and that the THERMALIZATION parameter is often misspelled and missing the second I. Also, one could still attempt an automatic convergence check, especially if the size of the statistical errors can be estimated, even if it might be less reliable. If the implemented convergence check can be activated for a CT-QMC solver and would work when the tolerance is set higher than the statistical error, it would be good to state that explcitly (or that it can not be used or would definitely not work if that is not the case).

In section 6.1.6, the specific Wannier90-format input is not provided. Not knowing precisely what needs to be specified, my first attempt to change the file from 6.1.5 was not accepted by DCore, but a much larger file with all zero combinations added was. To generate this, I modified a script provided in a DCore online tutorial for a similar three-dimensional model, but such a script (or directly the input) for this specific example was not provided there. It would be helpful if the authors provide the input in some way, e.g. by adding the examples from the paper as tutorials to the online documentation as well and providing the input files there (and adding an explicit reference to them in the paper). Also, how to get the spin moment data shown in Fig. 10 is not explained. A note mentioning where to get it (e.g. to use the terminal output of dcore if that is really the easiest way) should be added.

In section 6.2, there is one minor issue by itself in the missing hermitian conjugate part of the spin-flip and pair-hopping terms of eq. (17) in comparison to eq. (1) of reference 24. Having a quick look at it, this might actually be the 'correct' form of the interaction, but the reference cited here explicitly adds it so the difference should at least be explained for clarity. Another minor issue is the consistency between the examples: here, interaction = kanamori is specified explicitly in the parameters while that is not done in the previous example. Looking into the online reference manual, kanamori is the default setting, but if that is relied on in section 6.1 it would be good to make that as clear as possible and mention it there or the parameter should be added explicitly there as well. The same applies to the use of first the temperature parameter T in 6.1 and then the inverse temperature parameter beta in 6.2.

In section 6.3, there is a typo in the section title and I find it strange that the Kanamori interaction parameters differ slightly between the text (second sentence of 6.3.2) and the parameter snippet shown below it, especially considering that the version of the tutorial in the online manual has the same values as the text. Also, if the colorbar range in Fig. 13 were reduced, the quantitatively different areas of the plots might be easier to distinguish visually.

In my own attempts to reproduce the results using the ALPS CT-HYB solver, i.e. those of sections 6.2 and 6.3, I also noticed that the solver sometimes terminated early / crashed with a rather unhelpful error message, with the solver printing an exception message involving 'Acceptance_rate_global_shift' and DCore then printing another exception message failing to access the sign, which would less often occur when raising the timelimit parameter, and the perturbation orders before and after measurement would often not be as close as in the example output in 6.2.2; in fact, the one before measurement would sometimes be -nan. To me this seems like possibly a problem due to too little thermalization time, but is especially surprising as I was calculating this on a fairly new machine with, I would assume, not significantly worse single-core performance than whatever the authors used. A comment on such potential issues in the example and a better error message in such cases in the program should ideally be added or, if I made an obvious mistake, clearer instructions be given in the relevant part.

Requested changes

1- Please consider fixing some or all of the minor issues mentioned in my comments above, particularly those in the part on section 2.

---

## Round 2 · Referee Report · Anonymous (Referee 1) · 2021-1-20

Report

The authors have carefully taken into account most of my comments on the minor issues with the previous submission and I think that the clarifications and added details should help to avoid open questions or potential misunderstandings of this well written paper. I definitely recommend this improved version for publication in SciPost Physics.

Regarding the authors' comment on my attempt to reproduce the example in section 6.1.3 I can confirm that this works fine with the current version of DCore, and have now also found that it would have required manually setting a higher value of omega_pade with the version I used for my previous report.
  • validity: -
  • significance: -
  • originality: -
  • clarity: -
  • formatting: -
  • grammar: -

Author:  Hiroshi Shinaoka  on 2021-02-01  [id 1193]

(in reply to Report 1 on 2021-01-20)
Category:
answer to question

We thank you for reviewing the manuscript and the reply.
Regarding the comment in the second report,
the default value of omega_pade has been changed to a very high value in the latest version of DCore,
which seems to be a reasonable option.

---

## Round 2 · Referee Report · Hugo Strand (Referee 2) · 2021-2-4

Strengths

  1. Great open source project substantially lowering the unnecessary high bar for performing DFT+DMFT calculations.
  2. Great reference documentation.
  3. Great practical tutorials.

Weaknesses

  1. The introductory part of the manuscript is very accessible for experts in the field, but could potentially be challenging for the aspiring PhD student.

Report

Dear Editor,

Thank you for sending me the manuscript "DCore: Integrated DMFT software for correlated electrons" by Prof. Shinaoka et al. submitted for consideration in SciPost Physics.

In the manuscript the authors present the open source DCore project aimed at making electronic structure calculations for materials with strong local correlations using the combination of ab initio density functional theory (DFT) and the dynamical mean-field theory (DMFT) approximation. As pointed out in the manuscript DFT+DMFT is a well established method, however, compared with standard DFT calculations the entry bar for new practitioners has remained very high. This has, in part, to do with the plethora of algorithms employed to solve the DMFT many-body problem and their individual quirks and limitations. While, today, there are a number of established open source DMFT solvers, it remains non trivial for researchers entering the field to combine a DMFT solver with a DFT code and perform high quality DFT+DMFT calculations. The authors note that there are a few projects aiming at a complete DFT+DMFT compute platform (DMFTwDFT and eDMFT) which are either limited in their solver support or not open source. The DCore project aims to over come these limitations by supporting multiple solvers from different groups, e.g. the TRIQS and ALPS ecosystems, and by being open source and accessible to the research community world wide.

I find the manuscript well written and clear for practitioners in the field, however, I think that it could be made even more accessible for aspiring PhD students by explaining some of the DFT+DMFT specific concepts in more detail, please see specific comments below. I recommend the authors to take this into consideration. In my opinion the manuscript meets the expectation of SciPost Physics by "Presenting a breakthrough on a previously-identified and long-standing research stumbling block", namely, enabling non expert researchers to perform high quality DFT+DMFT calculations. Therefore I recommend the manuscript for publication in SciPost Physics.

Sincerest regards,
Hugo U.R. Strand

Requested changes

  1. Please define the TRIQS project abbreviation.
  2. Please cite the original TRIQS reference Ref. [24] on first mention of the project.
  3. The notion of charge self-consistency is only mentioned, not referenced or explained. Please consider adding references on the topic.
  4. The formulation in the Methodology section:

"The current version of DCore implements only one-shot DFT+DMFT calculations based on Wannier90 and does not support charge self-consistent calculations. In other words, it specializes in the analysis of multi-orbital Hubbard models, making it different from the programs eDMFT and DMFTwDFT."

can be misinterpreted as if the eDMFT and DMFTwDFT projects do not support analysis of multi-orbital Hubbard models. Please consider a reformulation.

  1. I recommend the authors to spend a paragraph on formulating the problem of the electron structure in a periodic system. Defining notions such as the unit-cell, spin-orbital indices, correlated and non-correlated shells, real space unit-cell coordinates, momentum space k-vectors, etc. before defining he Hamiltonian in the Model section Eqs. (1-2). I think this would make the manuscript more accessible for non-experts.

  2. After Eq. (3) the local interaction is limited to intra-shell interaction, I think it would be in place to stress that it is also assumed to be limited to local interaction in each unit-cell.

  3. For the uninitiated condensed matter theorist I think the statement

"All non-interacting orbitals (e.g., deep oxygen orbitals) belong to the non-interacting shell."

can be misleading. These electrons in the deep oxygen orbitals are -- in fact -- interacting with the Coulomb interaction, however the interaction is treated on a DFT level rather than DMFT. I recommend the authors to explain the different levels of approximations for the electron-electron interaction used in the correlated and (the in the DMFT treatment) non-interacting shells.

  1. Please consider name/define all quantities introduced in Eqs. (4-7) preferably before the equations.

  2. Please consider defining in what space the Green's functions and self-energies are matrices in before Eq. (8).

  3. The $\Sigma^{imp}$ self-energy is defined in Eq. (7) but not used in Eq. (8-9) please clarify the connection between impurity and lattice self energy.

  4. I find the statement

"The Green’s functions and the self-energies are assumed to be either spin-diagonal or spin-off-diagonal."

potentially confusing. It can be interpreted as G and Sigma can only be either purely diagonal or purely off-diagonal. Please consider clarifying the second case with probably? a dense matrix representation in spin-space with both diagonal and off-diagonal components.

  1. Since the mixing factor is called sigma_mix while $w$ is used in the manuscript in e.q. Eq. (11), please consider changing the manuscript as to agree with the DCore syntax and use $\sigma$ instead.

  2. In the list of double counting approaches, the "dressed" Green's function is mentioned. Unfortunately I can not find this Green's function defined in the manuscript. Please consider naming the quantities in Eq. (4-7) to increase clarity.

  3. I find the spin_orbit flag confusing and potentially misleading. Is its function equivalent to the DFT notion of "collinear" and "non-collinear" calculations? If yes, I recommend the authors to change the flag in DCore to adhere to the DFT community lingo.

  4. On page 8 the DFTTools project is mentioned but not cited, please consider adding a citation there.

  5. Please note that the indices $\alpha$ and $\beta$ are defined as spin-orbital indices in Eqs. (1-2). I think the definition of the interaction 4-rank tensor on page 8 for the Kanamori interaction breaks this, using the same indices for orbital only indices. Please consider clarifying this.

  6. References for interactions: Please add (original) references for the tree types of interactions, Kanamori, Slater-F and Slater-UJ interactions. In particular I think Table 2 deserves explanation/referencing.

  7. Please consider moving the http links, now embedded in the text, to the list of references.

  8. The current http links are pointing at the master-branch documentation. I.e. their content will change as the master-branch evolve. I recommend the authors to use a release branch/tag when linking to the documentation in order for the content to agree with the manuscript also in the future.

  9. Technical question: Why is DCore, being a pure python project, using CMake for installation? In my opinion using the Python ecosystem module installation approach enabling pip install would be a little more user-friendly.

  10. On page 13 the pomerol ED solver is mentioned, please consider adding a reference there.

  11. In Eq. (20-21) new second-quantized notation is introduced with $d$ operators and $\alpha$, $\beta$ as orbital-only indices, not in line with Eq. (1-2). Please consider making the notation coherent.

---

## Round 2 · Author Response

We thank you for your handling of our manuscript entitled "DCore: Integrated DMFT software for correlated electrons" and the referee for his/her very careful reading and a lot of feedback. The referee accepts the usefulness of the DFT+DMFT program (DCore) presented in the manuscript and is satisfied with the quality of the manuscript. The referee explicitly says ``both the presented program package as well as this paper are very nice work and that it is suitable for publication with at most minor revisions necessary''. The referee however provides many suggestions/comments for minor revisions (mainly for improving the readability) throughout the manuscript. We examined each suggestion/comment very carefully and made minor revisions throughout the manuscript. We reply to the comments one by one below.

Best regards, Hiroshi Shinaoka, Junya Otsuki, Mitsuaki Kawamura, Nayuta Takemori, Kazuyoshi Yoshimi

Comment on Section 3

Section 3 nicely lays out the structure of the package. The only minor issues I can possibly think of are that the figure suggests two different input files for dcore_pre and dcore while the text specifies a single text file, that one single arrow pointing from impurity solvers to dcore is probably not the best way to graphically represent that relationship (it is at least not the same as in the other cases where such an arrow is used, perhaps a double-headed arrow or two arrows would be better), and that the dcore_check program is missing from this section entirely for some reason.

Thank you for giving us useful comments for improving the readability. We've updated Figure 3 and added a description of dcore_check following the comments.

Comment on Section 4

Section 4.1 is a good summary of the available input parameters, but only the parameters of the model block are explained in detail. I think that explanations how the important parameters of the other blocks, especially system, should be used, possibly even just their names which are mostly missing now, would be helpful additions; if more detailed explanations are not given, at least a more explicit reference to the online reference manual should be added.

Thank you for pointing out the insufficient description for input parameters in blocks other than the model block. We have added an explicit reference to the online manual in each block. In addition, we have updated the description in the tool block.

Apart from that, minor issues are that the 'included blocks' sentence in the first paragraph, apart from the typo, could be expressed more clearly (i.e. that all programs usually read the same file, but that some blocks are, even if present, irrelevant to some of the programs) and that the image for the 'cubic' lattice seems a little visually cluttered and possibly confusing to me, partially also due to the inconsistency that the red cube does not indicate a unit cell while the red squares in the 'square' case do if I interpret the images correctly.

We have updated the first paragraph of Sec. 4.1 and Figure 4 following the comments.

Section 4.2 is a good summary as well. Points that I would clarify are what the 'kind' of correlated shell specifically refers to in the paragraph on dcore_pre and what quantity specifically is called the 'density of states' in this context in the paragraph on dcore_post. There are some potentially confusing typos here as well: the dcore paragraph incorrectly begins with 'dcore_pre' and the parenthesized 'renormalization factor' is placed between 'chemical' and 'potential' instead of after both above eq. (9).

Thank you for pointing out the confusing terminology. We have replaced density of states by spectral function and fixed the typo following the comments.

The installation section currently only refers to the online documentation. I think it would be better to give the installation commands at least for the simplest case (e.g. assuming some reasonably normal configuration, that the prerequisites are already installed and that no problems occur) in the paper as well, e.g. four lines for cloning the repository, calling cmake, make and make install or something short like that, so that a reader could follow the paper from beginning to end without having to consult much (or ideally any) additional material under ideal conditions.

Thank you for the suggestions. We provide typical building commands in the revised manuscript.

Comment on Section 6

Minor issues in section 6.1.2 are the in my opinion unclear explanation of the averaging influenced by sigma_mix (i.e. what exactly is averaged and when) and lack of an explanation of what the particular significance of its numerical value is, the slightly ambiguous phrasing of what exactly is executed in parallel at the bottom of the same page, and the description of the restart option, in particular what exactly it does, how it differs from just increasing max_step if it does, and, in that case, why it might be preferable to use it.

We have added, at the end of 2.2, an explanation for averaging influenced by sigma_mix with its explicit definition in Eq.(11). We have added more descriptions on MPI parallelization in Sec. 2.3 of the revised manuscript. The restart option is now explained in more detail in 6.1.2. If restart=True is activated, "then, dcore uses the final result for \Sigma(i\omega_n) in the previous run as the initial state and executes additional maxstep iterations. We note that the default is restart=False, which means the self-consistent loop is started over even though the previous result exists in the current directory."

Further, I find the advice to reduce sigma_mix if the convergence graph exhibits oscillations surprising; I would rather have expected the opposite here if anything, assuming that a higher numerical value of sigma_mix means more influence from earlier self-energy, but then again sigma_mix is not really explained in detail as mentioned. Also, considering that use by non-experts is a stated goal, I am not sure if it is very helpful to instruct the user to just enter some values of unclear origin like 0.5 or 1.0 for not well explained parameters like sigma_mix; I do not think that, having read this paper, a non-expert could be confident in the choice of parameters like sigma_mix. Advice like to change sigma_mix in case of oscillations is probably helpful in such cases (if correct), but advice on how to set it to begin with or specifying whether it is usually even necessary to set it manually would be a good addition.

The explicit definition for sigma_mix is now provided in Eq.(11). A larger numerical value of sigma_mix means more influence from new self-energy. We believe that the description is now clear.

For a guide for non-experts, we have added more introductory explanations for how to choose and change the value of sigma_mix. In the following, we quote the sentences which we have added below Eq.(17): "sigma_mix specifies the mixing parameter w in Eq. (11). The optimal value of sigma_mix depends on models and parameters. Generally, a larger value of sigma_mix (1 at the largest) can lead to a faster convergence, but may result in a bad convergence (e.g., oscillation and discontinuous change). A recommended strategy is that one starts from a large value such as 1and 0.5 and decreases it if the convergence graph (see below) does not show a tendency of convergence."

To the parameters of the CT-HYB-SEGMENT solver in 6.1.4 something similar applies, but if that solver is a special case (as the ALPS CT-HYB solver is later used with only a timelimit as configuration) then it would be good to point out explicitly that different solvers might require different amounts of configuration to get good results (and maybe also which solver should be used to avoid manual configuration as much as possible).

Thank you for the suggestion. We have added one paragraph at the end of Sec. 6.1.4 to clarify this point.

In section 6.1.3, the analytical continuation is glossed over, and it is not explained in depth anywhere else in the paper either. Considering the potential influence on the quality of the results, perhaps at least some short advice or a reference in case of problems in the users' own calculations might be appropriate.

We have added two paragraphs: The last paragraph in 6.1.3 mentions a possible unphysical result (negative spectrum) and its workaround, and the third last paragraph in 6.1.4 describes a possible uncertainty of analytical continuation when the self-energy involves statistical errors.

Further, in my own attempt to reproduce the results shown here, A(k, omega) looked significantly different from Fig. 7a. The artificial features that are explained to come from the discretization to three bath sites were not recognizable, instead even with n_bath = 3 and using the ED solver my results looked more like 7b than 7a. Maybe the authors can think of a reason for this, possibly I made a mistake somewhere, in that case the relevant step should probably be explained more clearly and otherwise the wrong part be fixed.

We have updated the online tutorial to be fully consistent with the manuscript. The exact input file is now provided there. We have confirmed in two different environments, that Fig.8(a) [Fig.7(a) in the previous manuscript] is indeed obtained.

We mention at the beginning of 6.1 that the input is available in the online tutorial.

For the plots similar to Fig. 7 it might be also helpful to note that to generate as nice plots as the authors show, the range of the colorbar might need to be restricted manually. The plot scripts generated by dcore_post do not restrict the range, which can result in a rather homogeneous look due to tiny outlier spots, as it happens in Fig. 13 for example.

We have added the sentence "We note that the range of the color bar needs to be changed manually to obtain a distinct spectrum." after Fig.8(a) is mentioned in 6.1.3.

The only minor issues with 6.1.4 are, as mentioned, that it might be hard for non-experts to know what values to choose for some of the CT-QMC solver parameters (N_MEAS is for example mentioned as an important parameter but not explained at all) and that the THERMALIZATION parameter is often misspelled and missing the second I.

Thank you for pointing this. We have added an explanation for N_MEAS in 6.1.4. Typos of THERMALIZATION has been fixed.

Also, one could still attempt an automatic convergence check, especially if the size of the statistical errors can be estimated, even if it might be less reliable. If the implemented convergence check can be activated for a CT-QMC solver and would work when the tolerance is set higher than the statistical error, it would be good to state that explcitly (or that it can not be used or would definitely not work if that is not the case).

That's right. One can use the automatic convergence check if the magnitude of statistical errors is known in advance. For better usage, we have added a new parameter n_converge, which controls the convergence check. The DMFT loop is terminated when the convergence criterion is satisfied n_converge times consecutively. This prevents the DMFT loop from being terminated prematurely.

We added a paragraph at the end of 6.1.4, and explain how to use the convergence check for QMC solvers. Thank you for bringing our attention to this.

In section 6.1.6, the specific Wannier90-format input is not provided. Not knowing precisely what needs to be specified, my first attempt to change the file from 6.1.5 was not accepted by DCore, but a much larger file with all zero combinations added was. To generate this, I modified a script provided in a DCore online tutorial for a similar three-dimensional model, but such a script (or directly the input) for this specific example was not provided there. It would be helpful if the authors provide the input in some way, e.g. by adding the examples from the paper as tutorials to the online documentation as well and providing the input files there (and adding an explicit reference to them in the paper).

We have updated the online tutorial and provide the full input files as well as a script for generating the Wannier90-format input. We have added a link to the online tutorial in the second paragraph of 6.1.6.

Also, how to get the spin moment data shown in Fig. 10 is not explained. A note mentioning where to get it (e.g. to use the terminal output of dcore if that is really the easiest way) should be added.

The spin moment data is easily obtained using a future implemented in the GitHub repository. We have merged it into the master branch. Fig.10 is now automatically generated by dcore_check. The online tutorial is also updated to be consistent with the manuscript.

We thank the referee for making us realize this issue.

In section 6.2, there is one minor issue by itself in the missing hermitian conjugate part of the spin-flip and pair-hopping terms of eq. (17) in comparison to eq. (1) of reference 24. Having a quick look at it, this might actually be the 'correct' form of the interaction, but the reference cited here explicitly adds it so the difference should at least be explained for clarity. Another minor issue is the consistency between the examples: here, interaction = kanamori is specified explicitly in the parameters while that is not done in the previous example. Looking into the online reference manual, kanamori is the default setting, but if that is relied on in section 6.1 it would be good to make that as clear as possible and mention it there or the parameter should be added explicitly there as well. The same applies to the use of first the temperature parameter T in 6.1 and then the inverse temperature parameter beta in 6.2.

Thank you for pointing out the inconsistency and typos. We have fixed the equation for the Hamiltonian and remove the parameter interaction for consistency with other examples. We have also added a description of the parameters T and beta.

In section 6.3, there is a typo in the section title and I find it strange that the Kanamori interaction parameters differ slightly between the text (second sentence of 6.3.2) and the parameter snippet shown below it, especially considering that the version of the tutorial in the online manual has the same values as the text. Also, if the colorbar range in Fig. 13 were reduced, the quantitatively different areas of the plots might be easier to distinguish visually.

Thank you for pointing out the typo. We have fixed the interaction parameters in the parameter snippet. We changed the color bar range of Fig. 14 and added the label for the spectrum.

In my own attempts to reproduce the results using the ALPS CT-HYB solver, i.e. those of sections 6.2 and 6.3, I also noticed that the solver sometimes terminated early / crashed with a rather unhelpful error message, with the solver printing an exception message involving 'Acceptance_rate_global_shift' and DCore then printing another exception message failing to access the sign, which would less often occur when raising the timelimit parameter, and the perturbation orders before and after measurement would often not be as close as in the example output in 6.2.2; in fact, the one before measurement would sometimes be -nan. To me this seems like possibly a problem due to too little thermalization time, but is especially surprising as I was calculating this on a fairly new machine with, I would assume, not significantly worse single-core performance than whatever the authors used. A comment on such potential issues in the example and a better error message in such cases in the program should ideally be added or, if I made an obvious mistake, clearer instructions be given in the relevant part.

Thank you for your efforts in reproducing the data and the inconvenience. We have to agree that controlling the results of QMC calculations is still difficult due to their sensitivity to the number of processors and simulation time. It seems that the QMC program runs very slowly in your environment due to some unknown reasons. Each QMC code may behave differently depending on the environment and thus it is difficult to provide a general solution. We have added one paragraph to warn the user of the instability of QMC and recommend the user to create an issue on GitHub if any problem.

---

## Round 2 · List of Changes

• Section 2: We have added Figure 1 and explained more technical details of DFT+DMFT calculations. We have added Sec. 2.3 on MPI parallelization and have removed the last subsection in Sec. 2.
  • Section 3: We have updated Figure 3 and added a description of the program dcore_pre following the suggestions.
  • Section 4: We have made minor updates for Figure 4 and updated the text following the suggestions.
  • Section 6.1: We have added more descriptions especially on technical details following the suggestions.
  • We have made minor revisions through the manuscript to improve the readability following the suggestions and comments.

---

## Round 3 · Author Response

We thank you for your handling of our manuscript entitled "DCore: Integrated DMFT software for correlated electrons" and the referee for his/her very careful reading and a lot of feedback. The referee accepts the usefulness/greatness of the DFT+DMFT program (DCore) presented in the manuscript and is satisfied with the quality of the manuscript. The referee explicitly says ``I find the manuscript well written and clear for practitioners in the field''. The referee however provides suggestions/comments for minor revisions for further improving the readability. We examined each suggestion/comment very carefully and made minor revisions throughout the manuscript. We provide a summary of major changes below. In addition, we highlight changes in the enclosed copy of the revised manuscript.

Best regards, Hiroshi Shinaoka, Junya Otsuki, Mitsuaki Kawamura, Nayuta Takemori, Kazuyoshi Yoshimi

---

## Round 3 · List of Changes

• Section 2: We have added Figure 1 and explained more technical details of DFT+DMFT calculations. We have added Sec. 2.3 on MPI parallelization and have removed the last subsection in Sec. 2.
  • Section 3: We have updated Figure 3 and added a description of the program dcore_pre following the suggestions.
  • Section 4: We have made minor updates for Figure 4 and updated the text following the suggestions.
  • Section 6.1: We have added more descriptions especially on technical details following the suggestions.
  • We have made minor revisions through the manuscript to improve the readability following the suggestions and comments.
  • We have updated DCore to version 3.0.0, which supports TRIQS 3.0.x and Python3.

Comment #1-#2

Thank you for pointing out the mistakes. We have updated the manuscript following the comments.

Comment #3-#4

Thank you for pointing out the misleading expression on the charge self-consistent calculations. We have reformulated these sentences as follows.

DCore does not support charge self-consistent calculations [9], which are supported by other programs such as eDMFT and DMFTwDFT. In a charge self-consistent calculation, the DFT effective potential is updated using the density matrix obtained by DMFT calculations. This is essential especially in performing lattice optimization. DCore specializes in the analysis of multi-orbital Hubbard models.

Comment #5

I recommend the authors to spend a paragraph on formulating the problem of the electron structure in a periodic system. Defining notions such as the unit-cell, spin-orbital indices, correlated and non-correlated shells, real space unit-cell coordinates, momentum space k-vectors, etc. before defining he Hamiltonian in the Model section Eqs. (1-2). I think this would make the manuscript more accessible for non-experts.

Thank you for the suggestion. We have added one paragraph at the beginning of Sec. 2.2 to define a lattice, primitive vectors, and Fourier transform.

Comment #6

After Eq. (3) the local interaction is limited to intra-shell interaction, I think it would be in place to stress that it is also assumed to be limited to local interaction in each unit-cell.

Thank you for the suggestion. We have modified the sentence as follows.

... is limited to intra-shell interactions and local interactions in each unit cell.

Comment #7

For the uninitiated condensed matter theorist I think the statement
"All non-interacting orbitals (e.g., deep oxygen orbitals) belong to the non-interacting shell." can be misleading. These electrons in the deep oxygen orbitals are -- in fact -- interacting with the Coulomb interaction, however the interaction is treated on a DFT level rather than DMFT. I recommend the authors to explain the different levels of approximations for the electron-electron interaction used in the correlated and (the in the DMFT treatment) non-interacting shells.

Thank you for pointing out the insufficient explanation of the concept of "non-interacting shell". In Sec. 2.1 of the revised manuscript, we explain the procedure of one-shot DFT+DMFT calculations and the meaning of ``intereacting'' at the DMFT level.

Comment #8--#9

8. Please consider name/define all quantities introduced in Eqs. (4-7) preferably before the equations. 9. Please consider defining in what space the Green's functions and self-energies are matrices in before Eq. (8).

Thank you for the suggestion. We have revised the first sentence of Sec. 2.3 to name all quantities in the equation and to explicitly state that they are defined in the spin-orbital space.

Comment #10

The $\Sigma^\mathrm{imp}$ self-energy is defined in Eq. (7) but not used in Eq. (8-9) please clarify the connection between impurity and lattice self energy.

Thank you for pointing out the unclear point. We have added one more equation to relate the two self-energies as Eq. (10) in the revised manuscript.

Comment #11

I find the statement "The Green’s functions and the self-energies are assumed to be either spin-diagonal or spin-off-diagonal." potentially confusing. It can be interpreted as G and Sigma can only be either purely diagonal or purely off-diagonal. Please consider clarifying the second case with probably? a dense matrix representation in spin-space with both diagonal and off-diagonal components.

Thank you for pointing out the misleading statement. We rephrased this sentence as follows.

The Green's functions and the self-energies are assumed to be either spin-diagonal or dense matrices in the spin space with both diagonal and off-diagonal components.

Comment #12

Since the mixing factor is called $\sigma_\mathrm{mix}$ while $w$ is used in the manuscript in e.q. Eq. (11), please consider changing the manuscript as to agree with the DCore syntax and use $\sigma$ instead.

Thank you for the suggestion. In the revised manuscript, we use $\sigma_\mathrm{mix}$ for the mixing parameter.

Comment #13

In the list of double counting approaches, the "dressed" Green's function is mentioned. Unfortunately I can not find this Green's function defined in the manuscript. Please consider naming the quantities in Eq. (4-7) to increase clarity.

Thank you for the suggestion. We have added a reference to Eq. (7) where this quantity is defined, and have renamed "dressed" to "local" in line with this equation.

Comment #14

I find the spin_orbit flag confusing and potentially misleading. Is its function equivalent to the DFT notion of "collinear" and "non-collinear" calculations? If yes, I recommend the authors to change the flag in DCore to adhere to the DFT community lingo.

Thank you for the suggestion and we appreciate it very much. However, the non-collinear option would be bit ambiguous because this flag could be used for non-magnetic calculations with spin-offdiagonal hoppings. Thus, we would like to consider introducing a new flag name in a future release based on feedback from the user.

Comment #15

On page 8 the DFTTools project is mentioned but not cited, please consider adding a citation there.

Thank you for pointing out the missing reference. We have added references to the paper of TRIQS/DFTTools wherever the DFTTools project is mentioned.

Comment #16

Please note that the indices $\alpha$ and $\beta$ are defined as spin-orbital indices in Eqs. (1-2). I think the definition of the interaction 4-rank tensor on page 8 for the Kanamori interaction breaks this, using the same indices for orbital only indices. Please consider clarifying this.

Think you for pointing out the inconsistency in our notations. On page 8 of the revised manuscript, we explicitly show the form of the kanamori interaction in the two notations to clarify the meaning of the indices.

Comment #17

References for interactions: Please add (original) references for the tree types of interactions, Kanamori, Slater-F and Slater-UJ interactions. In particular I think Table 2 deserves explanation/referencing.

Thank you for the suggestions. We have added references to an original paper and a review article in the caption of Table 2 and in the main text.

Comment #18

Please consider moving the http links, now embedded in the text, to the list of references.

Thank you for the suggestion. Following the suggestion, we have moved all the http links in the main text to the references.

Comment #19

The current http links are pointing at the master-branch documentation. I.e. their content will change as the master-branch evolve. I recommend the authors to use a release branch/tag when linking to the documentation in order for the content to agree with the manuscript also in the future.

Thank you for the suggestion. In the revised manuscript, the tag name, i.e. "v3.0.0", is used when linking to the online documentation.

Comment #20

Technical question: Why is DCore, being a pure python project, using CMake for installation? In my opinion using the Python ecosystem module installation approach would be a little more user-friendly.

Thank you for the technical suggestion on the build system. We have removed CMake from DCore. DCore v3.0.0 is now "pip-installable" as a pure Python package.

Comment #21

On page 13 the pomerol ED solver is mentioned, please consider adding a reference there.

Thank you for pointing out the missing reference. We have added a reference to pomerol where it is mentioned.

Comment #22

In Eq. (20-21) new second-quantized notation is introduced with $d$ operators and $\alpha$, $\beta$ as orbital-only indices, not in line with Eq. (1-2). Please consider making the notation coherent.

Thank you for pointing out the inconsistent notations. We have updated these two equations in line with Eqs. (1-2).

List of changes

  • Section 2: We have added Figure 1 and explained more technical details of DFT+DMFT calculations. We have added Sec. 2.3 on MPI parallelization and have removed the last subsection in Sec. 2.
  • Section 3: We have updated Figure 3 and added a description of the program dcore_pre following the suggestions.
  • Section 4: We have made minor updates for Figure 4 and updated the text following the suggestions.
  • Section 6.1: We have added more descriptions especially on technical details following the suggestions.
  • We have made minor revisions through the manuscript to improve the readability following the suggestions and comments.
  • We have updated DCore to version 3.0.0, which supports TRIQS 3.0.x and Python3.

---

## Editorial Decision

published